# A single pseudouridine on rRNA regulates ribosome structure and function in the mammalian parasite *Trypanosoma brucei*

K. Shanmugha Rajan [1,2], Hava Madmoni[1], Anat Bashan [2], Masato Taoka [3], Saurav Aryal[1], Yuko Nobe[3], Tirza Doniger [1], Beathrice Galili Kostin[1], Amit Blumberg[3], Smadar Cohen-Chalamish[1], Schraga Schwartz[4], Andre Rivalta [2], Ella Zimmerman[2], Ron Unger [1], Toshiaki Isobe[3], Ada Yonath[2] & Shulamit Michaeli [1] ✉

Trypanosomes are protozoan parasites that cycle between insect and mammalian hosts and are the causative agent of sleeping sickness. Here, we describe the changes of pseudouridine (Ψ) modification on rRNA in the two life stages of the parasite using four different genome-wide approaches. CRISPR-Cas9 knock-outs of all four snoRNAs guiding Ψ on helix 69 (H69) of the large rRNA subunit were lethal. A single knock-out of a snoRNA guiding Ψ530 on H69 altered the composition of the 80S monosome. These changes specifically affected the translation of only a subset of proteins. This study correlates a single site Ψ modification with changes in ribosomal protein stoichiometry, supported by a high-resolution cryo-EM structure. We propose that alteration in rRNA modifications could generate ribosomes preferentially translating state-beneficial proteins.

Eukaryotic ribosomes are highly conserved protein synthesis machines consisting of large and small subunit rRNA, 5.8S and 5S rRNA, and ~80 ribosomal proteins[1]. Until recently, ribosomes were considered a homogenous population, and translation regulation was attributed to translation factors and to proteins that bind specifically to mRNAs. In recent years, this dogma is being challenged, and evidence is accumulating to support the notion that the ribosome population is heterogenous, and that specialized ribosomes may exist that preferentially translate specific mRNA subsets[2]. Several factors can contribute to ribosome heterogeneity and specificity, such as rRNA sequence and modifications, and heterogeneity of ribosomal proteins (RP) (due to the existence of paralogs) as well as their stoichiometry and modification[3,4]. In mammals, changes in the translation of a subset of mRNAs are dependent on the presence of specific RPs, and their mutations or deregulation are associated with human diseases such as Diamond-Blackfan anemia[5] and/or cancer[2]. In yeast, the translation of

mitochondrial proteins is highly downregulated in cells lacking RP paralogs required for normal mitochondrial function[6].

rRNA modification can affect ribosome biogenesis and fine tune its specificity[7,8]. In mammals, approximately 50% of the modified sites are only partially modified[9] and in yeast, only a minority of modified sites vary across different conditions[10]. Dramatic induction of cytidine acetylation (ac⁴C) in ribosomes of an archeal hyperthermophile in response to increased growth temperature was observed[11]. The most abundant modifications on rRNA are 2′-*O* methylation (Nm) and pseudouridylation (Ψ)[9]. These modifications are guided by small nucleolar RNAs (snoRNA)[12–14]. The understanding of the role of Ψ modification was enhanced due to the development of genome-wide sequencing approaches. To date, three high-throughput sequencing technologies including Ψ-seq[15–17], HydraPsiSeq[18] and Oxford nanopore RNA sequencing[19] exist. Ψ-seq utilizes the preferential chemical modification of Ψ by N-Cyclohexyl-N′-(2-morpholinoethyl)carbodiimide

[1]The Mina and Everard Goodman Faculty of Life Sciences and Advanced and Nanotechnology Institute, Bar-Ilan University, Ramat-Gan 5290002, Israel. [2]Department of Chemical and Structural Biology, Weizmann Institute of Science, Rehovot 7610001, Israel. [3]Department of Chemistry, Graduate School of Science, Tokyo Metropolitan University, Minami-osawa 1-1, Hachioji-shi, Tokyo 192-0397, Japan. [4]Department of Molecular Genetics, Weizmann Institute of Science, Rehovot 76100, Israel. ✉e-mail: Shulamit.Michaeli@biu.ac.il

metho-p-toluenesulfonate (CMC), which causes the reverse transcriptase (RT) to stop one nucleotide before the modified base, resulting in decreased coverage of the modified nucleotide compared to the upstream one[15–17]. This method is semi-quantitative and sensitive to the surrounding RNA structure. Recently, a more quantitative method known as HydraPsiSeq was developed, based on hydrazine-aniline treatment, which cleaves only unmodified uridine, resulting in RNA fragments carrying Ψ that do not rely on RT stops[18]. The third method is Oxford nanopore RNA sequencing which relies on signatures of base-calling errors[19]. Finally, the tandem LC-MS method, based on precise mass and fragmentation information of RNA, provides accurate modification information, although its throughput is limited[9]. Recently, we have shown that to confidently determine the existence of rRNA modification and its level, more than a single high-throughput mapping approach is needed[20].

Impaired rRNA pseudouridylation in humans by mutations in the *dkc1* gene, which encodes the snoRNA associated pseudouridine synthase, affects the translation of a subset of mRNAs[21]. In yeast, global translation and fidelity are impaired when groups of Ψ modifications on helix 69 (H69) and the decoding center are altered[22,23]. However, no effect was observed following changes of a single modification[22,23]. The effect of Ψ modifications on H69 was studied in bacteria and yeast[22,24,25]. H69 interacts with the small subunit rRNA helix 44 (h44) just below the A-site, forming the ribosomal inter-subunit bridge (bridge 2a)[22,24,25]. Ψs on the H69 stabilize the helix, enabling the conformational changes of its loop during the dynamic translation process[22,24,25]. Most recently, based on the cryo-EM structure of Ψ-free yeast ribosomes, it was proposed that Ψs on H69 promote the interaction with h44, which may facilitate the small subunit head swivel during translation[26].

Trypanosomatids are parasites threatening the lives of ~70 million people worldwide. These parasites are known for the unique mechanisms governing gene expression such as *trans*-splicing[27] and mitochondrial RNA editing[28]. In the absence of transcriptional regulation for protein coding genes, gene expression is post-transcriptionally regulated mainly via mRNA stability and translation[29]. The parasite cycles between two hosts, insect and mammalian, and requires adaptation to temperature changes of up to 10 °C. Significant differences exist in metabolism, gene expression, and morphology between the *Trypanosoma brucei* (*T. brucei*) procyclic form (PCF), which propagates in the tsetse fly, and the bloodstream form (BSF), which replicates in the mammalian host[30]. We previously demonstrated that these parasites have a rich repertoire of 99 Nm and 68 Ψ on their rRNA[20,31]. The snoRNAs guiding these modification are encoded either by clusters of C/D and H/ACA snoRNAs that guide Nm and Ψ, respectively, or by solitary snoRNAs[32,33]. Overexpression of snoRNAs guiding modifications on rRNA H69 provided a growth advantage to PCF parasites at elevated temperatures[31]. Thus, hypermodified positions may contribute to the adaptation of ribosome function during cycling between the two hosts.

In this study, we revisited the mapping of Ψ levels in the two life stages using quantitative methods, namely HydraPsiSeq, direct RNA Oxford nanopore sequencing, and tandem LC-MS, which revealed the presence of hyper- and hypo-modified positions. Complete knock-out by CRISPR-Cas9 of the snoRNAs guiding Ψs on H69 was lethal, and a single knock-out (sKO) of a specific snoRNA affected growth and the translation of a subset of proteins. This sKO which is specific to Ψ530 affected the ribosome's structure, resulting in the dislodging of RP eS12 located on the surface of the ribosome. The lack of eS12 in ribosomes was confirmed by determining the 2.47 Å cryo-EM structure of the sKO *T. brucei* ribosomes. This study links a single change in rRNA Ψ modification with RP composition known to affect the translation of a subset of proteins[34]. Thus, alteration in rRNA modification during development and the disease state could generate ribosomes preferentially translating state-specific proteins.

## Results

### HydraPsiSeq, native RNA nanopore mapping, and tandem LC-MS on rRNA reveals both hyper- and hypomodified Ψ positions in the two life stages of *T. brucei*

Using Ψ-seq, we previously mapped Ψs on rRNA in the two life stages of the parasites and described hypermodified sites across the rRNA in functional domains[31]. Recently, a more quantitative method for mapping of Ψ, known as HydraPsiSeq was developed. This method is based on random cleavage of uridine nucleotides (not Ψ) in RNA, utilizing the combination of hydrazine treatment, which opens the ring structure of the pyrimidine base, and aniline treatment, which releases a 5'-phosphorylated RNA fragment[18]. Using this method, we calculated the Psi-Score, which estimates the fraction of uridine that is pseudouridylated. To validate this approach, we mapped the rRNA Ψ in the two life stages of the parasite. To examine the reliability of the method, we also mapped Ψ on rRNA derived from cells silenced for the H/ACA snoRNA-associated pseudouridine synthase, CBF5. The result (Supplementary Data 1) verified the existence of 70 Ψs, out of 74 sites that were previously detected by Ψ-seq[31]. Among the four undetected sites, two sites are also modified by an Nm modification. The level of modification on 66 Ψ sites was reduced following *cbf5* silencing. Comparison of Psi-Score of BSF versus PCF forms revealed the presence of three hyper-modified sites (SSU_Ψ2116, SSU_Ψ2146, LSUα_Ψ1481) and four hypo-modified sites (LSUα_Ψ1609, LSUβ_Ψ524, LSUβ_Ψ522, LSUβ_Ψ1419) in total RNA ($p$-value < 0.05) (Fig. 1a, Supplementary Data 2). Notably, all four hyper-modified and hypo-modified (LSUα_Ψ1609) sites in BSF were previously detected by Ψ-seq (Fig. 1b)[31].

Interestingly, two of the hyper-modified sites at Ψ522 and Ψ524 that were suggested by Ψ-seq to be hypermodified[31], were found to be hypo-modified by HydraPsiSeq. To further validate the stoichiometry of the Ψs, we used the direct RNA Oxford nanopore sequencing method[19] (Fig. 1b, Supplementary Data 3, Supplementary Fig. 1). This method (Supplementary Data 3) verified the existence of three hyper-modified sites (SSU_Ψ2116, SSU_Ψ2146, and LSUα_Ψ1481) when comparing BSF to PCF but failed to confirm the effect on Ψ522 and Ψ524 due to an unknown reason. Note, that the four Ψ sites that were not reduced following *cbf5* silencing by HydraPsiSeq (Supplementary Data 1) were shown to be reduced upon analysis with Oxford nanopore sequencing (Supplementary Data 3), suggesting that these are also guided by snoRNAs.

To obtain a sequencing-independent quantification of Ψ sites, we used tandem LC-MS[9,35]. LC-MS of total RNA derived from BSF and PCF cells verified the existence of all three hyper-modified sites (SSU_Ψ2116, SSU_Ψ2146, LSUα_Ψ1481) and one hypo-modified site (LSUα_Ψ1609) that was detected by HydraPsiSeq (Fig. 1b, Supplementary Data 4). It is currently not known why the three hypo-modified sites identified by HydraPsiSeq, LSUβ_Ψ524, LSUβ_Ψ522, and LSUβ_Ψ1419, could not be verified by other methods. Similarly, SSU_Ψ1619 was identified as a hypermodified site by Ψ-seq, nanopore sequencing and LC-MS, but not by HydraPsiSeq, while SSU_Ψ1088 was identified as a hypermodified site only by nanopore sequencing and LC-MS. All these sites (LSUβ_Ψ524, LSUβ_Ψ522, LSUβ_Ψ1419, SSU_Ψ1619 and SSU_Ψ1088) require further validation before they can be unequivocally assigned as differentially modified sites. Among the three hypermodified sites that were identified by at least three Ψ mapping methods, including HydraPsiseq, Ψ2146 is located in the small ribosomal subunit decoding center. Among the hypo-modified sites, Ψ1609 is located next to the peptidyl transferase center (PTC) and the polypeptide tunnel entrance regions. These sites are depicted on the cryo-EM structure of the trypanosomatid ribosome[36] (Fig. 1c).

We further sought to investigate whether the level of rRNA modification differs between total rRNA, and rRNA extracted from translating polysomes. To this end, polysomes were enriched by sucrose gradient fractionation from PCF and BSF extracts, and rRNA was subjected to HydraPsiSeq (Supplementary Data 2). The results

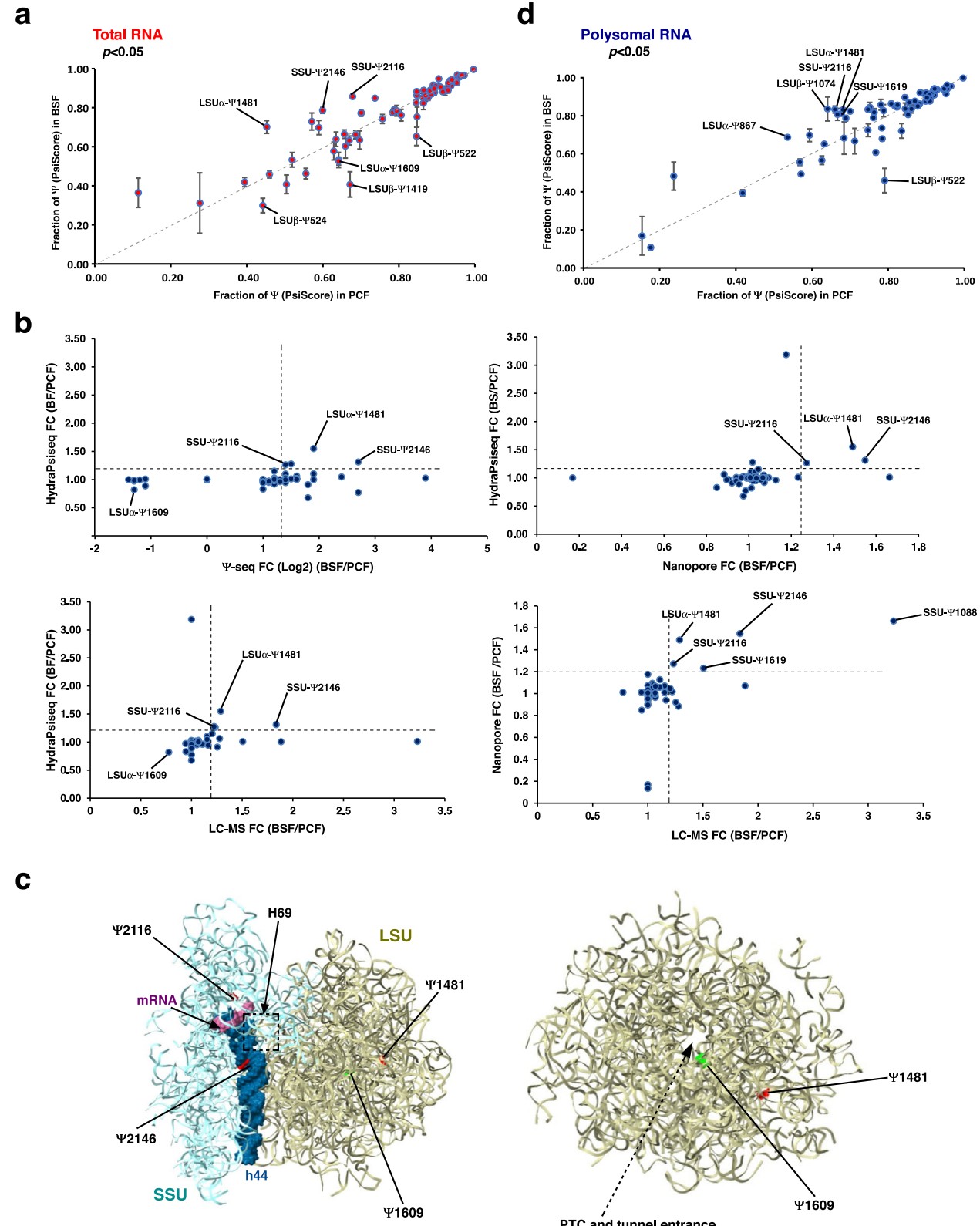

suggest that the translating polysomal rRNA has a higher fraction of Ψ compared to total rRNA (Supplementary Data 2). Comparison of Psi-Score from polysomal rRNA derived from PCF versus cultured BSF revealed five hypermodified (SSU_Ψ1619, SSU_Ψ2116, LSUα_ Ψ867, LSUα_ Ψ1481, LSUβ_Ψ1074), and a single hypo-modified site (LSUβ_Ψ522). We observed that SSU_Ψ2116 and LSUα_Ψ1481 are hyper-modified in both total and polysomal rRNA. Similarly, LSUβ_Ψ522 is

hypo-modified in both rRNA fractions, showing that only specific Ψ sites altered in total rRNA are incorporated in the translating ribo-somes. Moreover, seven Ψ sites in PCF and nine Ψ sites in BSF had higher Ψ stoichiometry in polysomal rRNA compared to total rRNA (Supplementary Data 2, Supplementary Fig. 2i), suggesting the pre-sence of a quality control mechanism. Interestingly, three of these variable sites are located in the H69 structure (Supplementary Fig. 2ii).

**Fig. 1 | Ψs on *T. brucei* rRNA are developmentally regulated. a** HydraPsiSeq analysis on RNA derived from total cell lysate. PsiScore (fraction pseudouridylated) is presented across the rRNA molecules, and the identity of significantly altered (*p* < 0.05) Ψ sites is indicated. Data are presented as mean ± S.E.M. Three biological replicates (*n* = 3) were used for the quantification. *p*-value was determined by Student's *t*-test Two-tailed distribution. No adjustments were made for multiple comparisons. The exact *p*-values of each data points are provided in Supplementary Data 2. **b** Comparison of differentially modified Ψ sites using four different mapping methods. The identities of the compared methods are indicated in the *x*- and *y*-axis. Ψ sites whose levels are altered similarly in both of the compared methods are indicated. **c** Localization of developmentally regulated Ψ sites on the trypanosomatid ribosome. Differentially regulated Ψ sites in *T. brucei* total rRNA are shown on the 3D structure of the ribosome[36]. Ψ sites whose levels are increased in BSF (compared to PCF) are depicted in red, and those whose levels are decreased are shown in green. **d** HydraPsiSeq analysis on RNA derived from polysomes. Psi-Score (fraction pseudouridylated) is presented across the rRNA molecules, and the identity of significantly altered (*p* < 0.05) Ψ sites is indicated. Data are presented as mean ± S.E.M. Three biological replicates (*n* = 3) were used for the quantification. *p*-value was determined by Student's *t*-test Two-tailed distribution. No adjustments were made for multiple comparisons. The exact *p*-values of each data points are provided in Supplementary Data 2. Source data are provided as a Source Data file.

## CRISPR-Cas9 single knock-out of the TB11Cs6H1 snoRNA guiding Ψ530 on H69 affects growth and translation in a specific manner

H69 harbors five Ψs in close proximity (Supplementary Data 2) whose location is depicted in the 3D structure of the trypanosomatid ribosome (Fig. 2a). Studies on ribosomes from prokaryotes and eukaryotes highlighted the importance of this rRNA helix for ribosome function, and for its dynamic movement during translation[22,24–26]. To understand the functional importance of these Ψs to the trypanosome ribosome, especially since some of the Ψs are regulated, we sought to manipulate the level of snoRNA guiding these Ψs. We made several attempts to knock-out all the snoRNA guiding Ψs on H69, namely TB9Cs2H1 (guiding Ψ518), TB7Cs1H1 (guiding Ψ522), TB9Cs5H1 (guiding Ψ524), TB3Cs2H1 (guiding Ψ528), and TB11Cs6H1 (guiding Ψ530) by CRISPR-Cas9, but all attempts failed except the sKO of TB11Cs6H1 and TB7Cs1H1, suggesting that even sKO of the other snoRNAs is lethal.

TB11Cs6H1, like all of the snoRNAs guiding Ψ on H69, is encoded by a solitary gene encoding a single snoRNA[33]. To achieve knock-out of the snoRNA gene, PCF cells constitutively expressing spCas9 from the PFR locus were used[37]. Two guide RNAs were used to target the spCas9 to the flanking sequence of the snoRNA. To select for the KO cells, we used a homologous DNA repair template (HDR) carrying the G418 resistance gene and tagRFPt flanked by 30 nucleotides upstream and downstream to the cleavage sites to drive the homologous integration[38] (Fig. 2bi). The HDR PCR product was transfected along with the in vitro synthesized gRNA, and transfected parasites were enriched by FACS and selected for G418 resistance. Among the selected G418 positive clones, only clones in which the level of snoRNA was reduced by ~50% were identified, suggesting that double KO is most likely lethal (Fig. 2bii, Supplementary Fig. 3). The proper integration was examined by PCR using primers located in the insert, and upstream to snoRNA (Supplementary Fig. 3b). The level of Ψ guided by the snoRNA was determined by HydraPsiSeq[18]. Reduction in the level of Ψ530 was observed, with no changes of Ψ at the neighboring positions (Fig. 2c). The reduction in the level of the snoRNA and Ψ affected growth, as can be seen by comparing the growth rate of three independent clones in comparison to the parental strain (PS) expressing the Cas9 protein (Fig. 2d).

Next, the effect on global translation was examined by following incorporation of methionine, using a commercial protein synthesis fluorescence assay. The results demonstrated a slight decrease in methionine incorporation (Fig. 2e). To determine whether the change in a single Ψ affected the translation of only a subset of proteins, we first assessed if any changes took place at the mRNA level. To this end, we performed RNA-seq of poly A+ selected RNA of the PS and compared it to the sKO cells. The library detected ~8900 mRNAs, but no significant change (*p* < 0.05) in mRNA level was observed in the sKO (Fig. 2f). Next, we examined changes in the proteome using isotope dimethyl labeling. This method is quantitative, does not require the in vivo incorporation of the isotope, and is fast and efficient[39]. Using this method, we quantified 2,468 proteins, and based on three replicates (fold change of 1.4 and *p*-value < 0.05) revealed 29 proteins that were reduced, and 48 that were elevated, in the sKO cell compared to the PS (Fig. 2g, Supplementary Data 5). Notably, the level of mRNA encoding the de-regulated proteins was unchanged.

Based on the proteome analysis, one of the proteins with reduced expression in the sKO was aldolase (ALD). To further examine the mechanism of this effect on protein levels, the 3′ untranslated region (UTR) of the aldolase gene was cloned into a dicistronic construct (described in Materials and Methods) (Fig. 2hi) and transfected to the PS and to TB11Cs6H1 sKO cells. The results (Fig. 2hii) demonstrate reduction in the level of BirA-HA carrying the aldolase 3′ UTR, suggesting that the effect on aldolase translation lies in the 3′ UTR, possibly by affecting the interactions of mRNA with the pre-initiation complex. Since the proteome detected only ~2468 proteins, we attempted to identify additional regulated proteins. To this end, sKO extracts were subjected to western analysis with antibodies specific to RNA binding proteins, generated by our group (Fig. 2i). We observed a reduction in the level of hnRNPF/H in all sKO clones, and not in the other tested proteins (Fig. 2i). Thus, the level of hnRNPF/H can be used as read-out for the effect of sKO on translation. Interestingly, three out of the seven proteins that were examined by western blot analysis do not appear in the proteome due to their low abundance in PCF, including hnRNPF/H. To investigate the mechanism by which the level of hnRNPF/H was reduced in the sKO cells, the authentic 3′ UTR of the gene was replaced by tubulin 3′ UTR in situ (Fig. 2ji) and the level of the tagged protein was monitored by western analysis (Fig. 2jiii). The results (Fig. 2j) demonstrate that as in aldolase, the effect on hnRNPF/H protein level lies in the 3′ UTR.

To search for other features potentially shared between mRNAs encoding proteins that are de-regulated in the sKO cells, we examined these mRNAs and found that 60 out of 77 proteins are developmentally regulated in the two life stages of the parasite, based on TriTrypDB (https://tritrypdb.org) annotation (Supplementary Data 6)[39]. We could not find any codon-usage differences in the ORF of proteins affected by sKO (Supplementary Fig. 4). Such codon usage bias was previously shown to affect translation under perturbation of rRNA modification[40]. However, as described above, the 3′ UTR of both aldolase and hnRNPF/H determines the de-regulation in the sKO cells. Further studies are needed to better understand how the interaction of the 3′UTR and/or its associated proteins mediate this regulation.

## Addback of wild-type TB11Cs6H1 reverses the sKO phenotype

sKO by CRISPR-Cas9 may have non-specific effects. Thus, to verify that the observed changes at the protein level (Fig. 2i) result from the absence of a single Ψ modification on rRNA that impacts translation, we restored the snoRNA level in the sKO by expressing the snoRNA from the tubulin locus[38]. In parallel, we also generated a mutated snoRNA unable to guide Ψ as a result of two mismatches in the guiding pocket (Fig. 3a). Both snoRNA overexpression constructs were transfected into the sKO strain. Northern analysis (Fig. 3bi) demonstrated the elevation of TB11Cs6H1 snoRNA in the sKO cells due to adding back either the wild-type or the mutated snoRNA (Fig. 3bii). The Ψ530 modification was recovered only in the wild-type snoRNA addback (Fig. 3c). The addback of the wild-type snoRNA but not of the mutated one was able to reverse the growth inhibition phenotype of the sKO

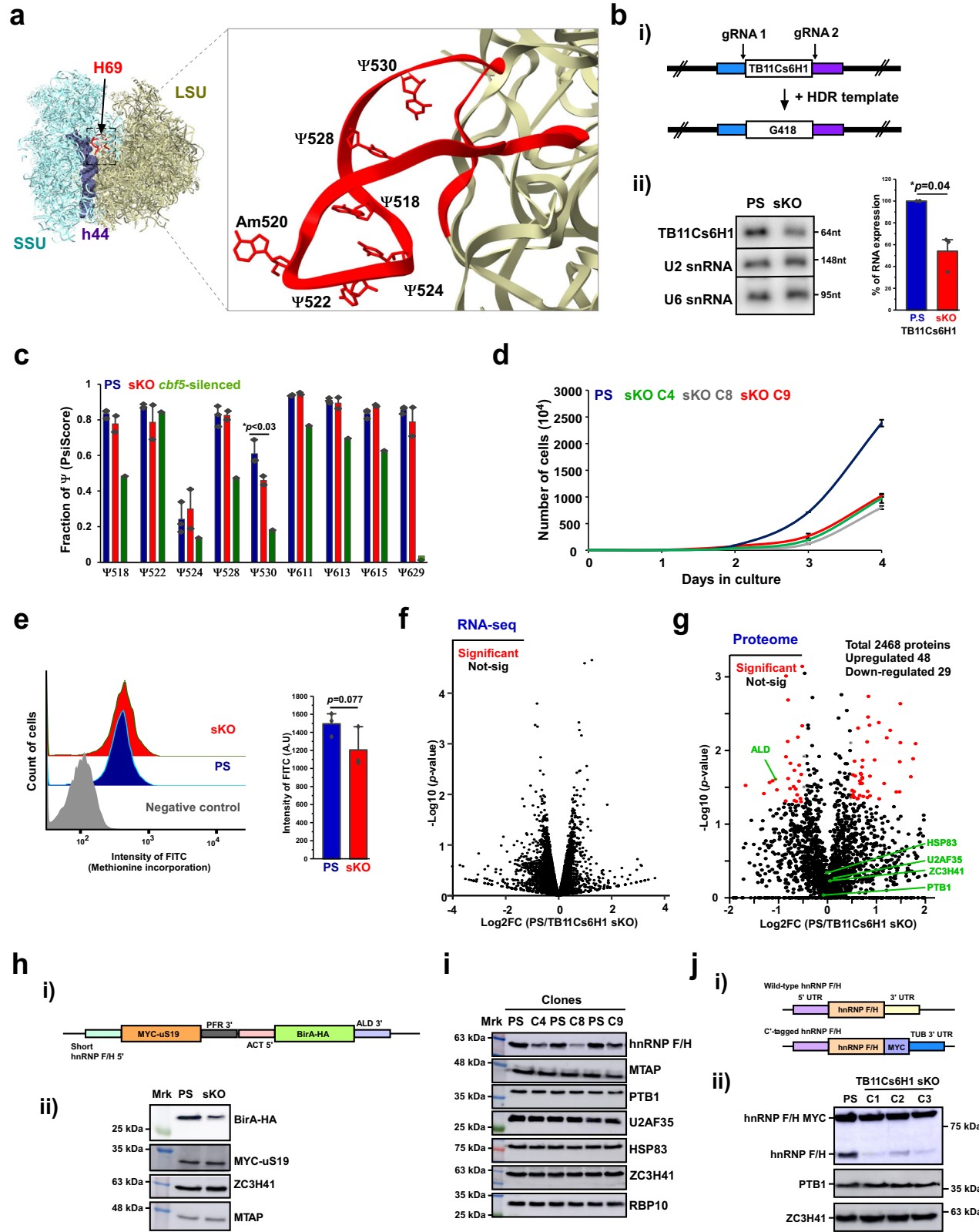

strain (Fig. 3d), suggesting that the change in the level of modification affected translation, and was not due to the possible chaperone function of the snoRNA.

To further validate that the observed phenotype correlates with the translation of hnRNPF/H with TB11Cs6H1 snoRNA addback, TB11Cs6H1 expression was induced from the rRNA locus under a Tet inducible EP procyclin promoter[41] (Supplementary Fig. 5a). Indeed,

wild-type snoRNA overexpression increased along the induction period, and the level of hnRNPF/H was increased accordingly, suggesting that translation of this protein depends on the level of Ψ530 (Supplementary Fig. 5b).

Next, we examined if sKO of different snoRNA species guiding on H69 affects translation similarly. The sKO of TB7Cs1H1 guiding Ψ522 that we prepared (Supplementary Fig. 6a, b), exhibited a growth defect

**Fig. 2 | TB11Cs6H1 snoRNA guiding Ψ530 on *T. brucei* LSU rRNA regulates translation. a** Localization of Ψ sites on the H69 domain of the trypanosome ribosome. **b** CRISPR-Cas9 knockout of TB11Cs6H1. (i) Schematic representation indicating the target of two guide RNAs and the site of HDR template integration. (ii) Validation of TB11Cs6H1 snoRNA single KO (sKO). Data are presented as mean ± S.D. Experiments were done in triplicate (*n* = 3). *p*-value was determined by Student's *t*-test Two-tailed distribution. \**p* = 0.04. **c** TB11Cs6H1 snoRNA guides Ψ 530 on the H69 domain. Three replicates of PS, two replicates of sKO and one replicate of cbf5-silenced RNA was used for HydraPsiSeq analysis. Data are presented as mean ± S.D. *p*-value was determined by Student's *t*-test Two-tailed distribution. \**p* = 0.03. **d** Growth of cells following TB11Cs6H1 sKO. Data are presented as mean ± S.E.M. Experiments were done in triplicate (*n* = 3). **e** Methionine incorporation assay. Experiments were done in triplicate (*n* = 3). Data are presented as mean ± S.D. *p*-value was determined by Student's *t*-test Two-tailed distribution. *p* = 0.077. **f** Transcriptome of TB11Cs6H1 sKO cells. Three biological replicates were

used to calculate the fold-change (FC). Corrected *p*-value and fold-change was calculated using DESeq2. The *p*-values were corrected for multiple testing using the Benjamini and Hochberg method. **g** Quantitative proteome of TB11Cs6H1 sKO cells. Three biological replicates were used to calculate the fold-change (FC) upon sKO. *p*-value was determined by one-sample *t*-test. Benjamini-Hochberg correction for multiple hypothesis testing (SignificanceB) was performed. **h** Validation of aldolase translation upon sKO. (i) Schematic representation of the dicistronic reporter cassette. (ii) Western analysis. ZC3H41 and MTAP served as loading controls. **i** Western analysis was done on three independent clonal populations of sKO cells. HSP83 served as loading controls. **j** hnRNP F/H translation upon sKO is dependent on its 3′ UTR. (i) Schematic representation of the UTR replacement strategy by in situ tagging with the tubulin (TUB) 3′ UTR. (ii) Three clones (C1-3) of sKO cells expressing MYC tagged hnRNPF/H are shown. All samples shown in Figures hii, i and jii were derived from the same experiment and blots were processed in parallel. Source data are provided as a Source Data file.

(Supplementary Fig. 6ci). An addback was obtained using the Tet-inducible promoter (Supplementary Fig. 6cii). Surprisingly, the sKO of TB7Cs1Hl as opposed to sKO of TB11Cs6H1 did not affect the level of hnRNP F/H, suggesting that the various Ψs on H69 differentially affect translation (Supplementary Fig. 6d).

## 80S monosome composition is altered in sKO of TB11Cs6H1 snoRNA

Next, we sought to examine the mechanism underlying the translation defect observed when the level of a single snoRNA is altered. The selective effect observed on translation due to reduction in the level of Ψ530 resembles the effect observed following changes in the levels of single ribosomal proteins in mammalian cells[34,42]. We therefore examined the impact of the sKO and reduction in the level of Ψ530 on 80S monosome composition (Fig. 4a). Since monosomes are easier to obtain in large quantities compared to polysomes, and because the sKO cells are growth inhibited, limiting the ability to obtain large amounts of polysomes, we examined monosome composition. Indeed, quantitative proteome analysis of 80S monosomes revealed a substantial reduction in the level of eS12 in sKO compared to the PS (Fig. 4b, Supplementary Data 7). The location of eS12 is at the head of the small subunit at the ribosome surface facing the solvent (Fig. 4c). To investigate whether the effect on translation stems from loss of eS12, the translation of hnRNPF/H protein was examined following overexpression of eS12 protein in the sKO cells. To this end, eS12 was tagged with the MYC tag at the N-terminus and was expressed from the tubulin locus[38]. The existence of MYC-eS12 in the ribosome was confirmed by sucrose gradient fractionation (Fig. 4di). Efficient expression of the tagged protein was achieved in both the PS and sKO strains (Fig. 4dii). We then quantified the expression of eS12 by mass spectrometry of 80S ribosomes that were visualized by negative staining. The results revealed 2.67-fold increase in the level of eS12 compared to sKO ribosomes (Supplementary Fig. 7). The ectopic expression of eS12 was able to abrogate the inhibition in hnRNPF/H translation in the sKO, suggesting that the reduction in this ribosomal protein contributes to the changes seen upon Ψ530 depletion (Fig. 4dii–iii). Ectopic eS12 overexpression in PS had a specific effect on protein synthesis, as shown by its effect on the synthesis of hnRPF/H, suggesting that the translation of hnRNPF/H is sensitive to variations in the quantities of this ribosomal protein (Fig. 4dii–iii). Next, we compared the proteome of the sKO cells to that of sKO expressing eS12 addback, and indeed, the synthesis of several proteins was recovered in an eS12 dependent manner (Supplementary Data 8). Although hnRNP F/H translation was restored, not all proteins behaved similarly upon eS12 addback, suggesting that not all translational defects can be attributed to the malfunctioning ribosomes (see Discussion).

The presence of ribosomes lacking eS12 was surprising, given the fact that such depletion is lethal in yeast and humans[43,44]. To determine the effect of eS12 silencing, the eS12 mRNA was silenced using a stem-

loop RNAi construct (Fig. 4e) and the silencing was confirmed by western blot analysis using in-situ MYC-tagged eS12 (Fig. 4ei). The growth of cells was compromised following eS12 silencing (Fig. 4eii), but unlike other eukaryotes[43,44] loss of eS12 did not induce any defects on 40S, 80S and polysome formation (Fig. 4f), suggesting that in trypanosomes, active ribosomes can be formed without eS12. To investigate the role of the 3′ UTR in translation regulation, we examined whether the expression of a BirA-HA reporter gene carrying aldolase 3′ UTR is altered upon eS12 silencing. The results demonstrate that the expression of the reporter carrying aldolase 3′ UTR depends on the presence of eS12 (Fig. 4g). These results support the notion that de-regulation of translation in the sKO depends on both the 3′ UTR and on the presence of eS12.

## High-resolution cryo-EM structure of *T.brucei* 80S ribosomes lacking eS12

To examine if the ribosome population in the TB11Cs6H1 sKO cells indeed lacks eS12, the structure of *T. brucei* 80S ribosomes was determined by cryo-EM. Accordingly, 80S ribosomes were purified from PS and TB11Cs6H1 sKO cells, their composition was verified by MS, and the purified ribosomes were subjected to cryo-EM data collection. MS analysis of these purified ribosomes suggested that eS12 was absent in sKO ribosomes. The initial cryoEM dataset included 459985 and 552813 particles for the PS and sKO ribosomes, respectively, yielding a 2.93 and 2.95 Å reconstruction of their entire 80S structure, respectively (Supplementary Fig. 8). To improve the resolution of the EM map, several rounds of 3D classification, Bayesian polishing, and refinement were performed. The final EM map of PS and sKO had an improved resolution of 2.47 Å (PDB ID: 8OVA) and 2.6 Å (PDB ID: 8OVE), respectively (Supplementary Fig. 8). Upon multibody refinement, the map quality of head and body regions of the small subunit (SSU) were further improved (Supplementary Fig. 8). The resulting EM map allowed us to clearly visualize the entire H69 in both maps. These high-resolution EM maps allowed us to build, refine and model all 9 rRNA fragments (SSU, the two large subunit (LSU) fragments, 5.8S, srRNA 1, 2, 4, 6, 5S), E-site tRNA, and 42 and 34 ribosomal proteins in the LSU and SSU, respectively (Table 1, Fig. 5a, Supplementary Data 9). Thus, the EM map we attained enabled us to correct and update the previously published 5.57 Å low-resolution structure of *T. brucei* ribosomes[45]. Using these maps, we could visualize and model 82 Nm, 6 base modifications including m$^1$acp$^3$Ψ, m$^6_2$A, m$^5$C, m$^7$G, and 43 Ψ residues (Fig. 5b, Supplementary Data 10). The modeled Ψ positions are in concurrence with Ψ-seq and HydraPsiSeq detection (Supplementary Data 10). Interestingly, four conserved modifications (Ψ942, Ψ996, Ψ1747 and Am254) were localized to trypanosomatid-specific expansion segments (Supplementary Fig. 9a). Our cryo-EM structures also reveal two Nm modifications (Um7 and Am163) near the focal points of the ribosome, that are the domains at which the termini of the rRNA fragments coincide[46](Supplementary Fig. 9b).

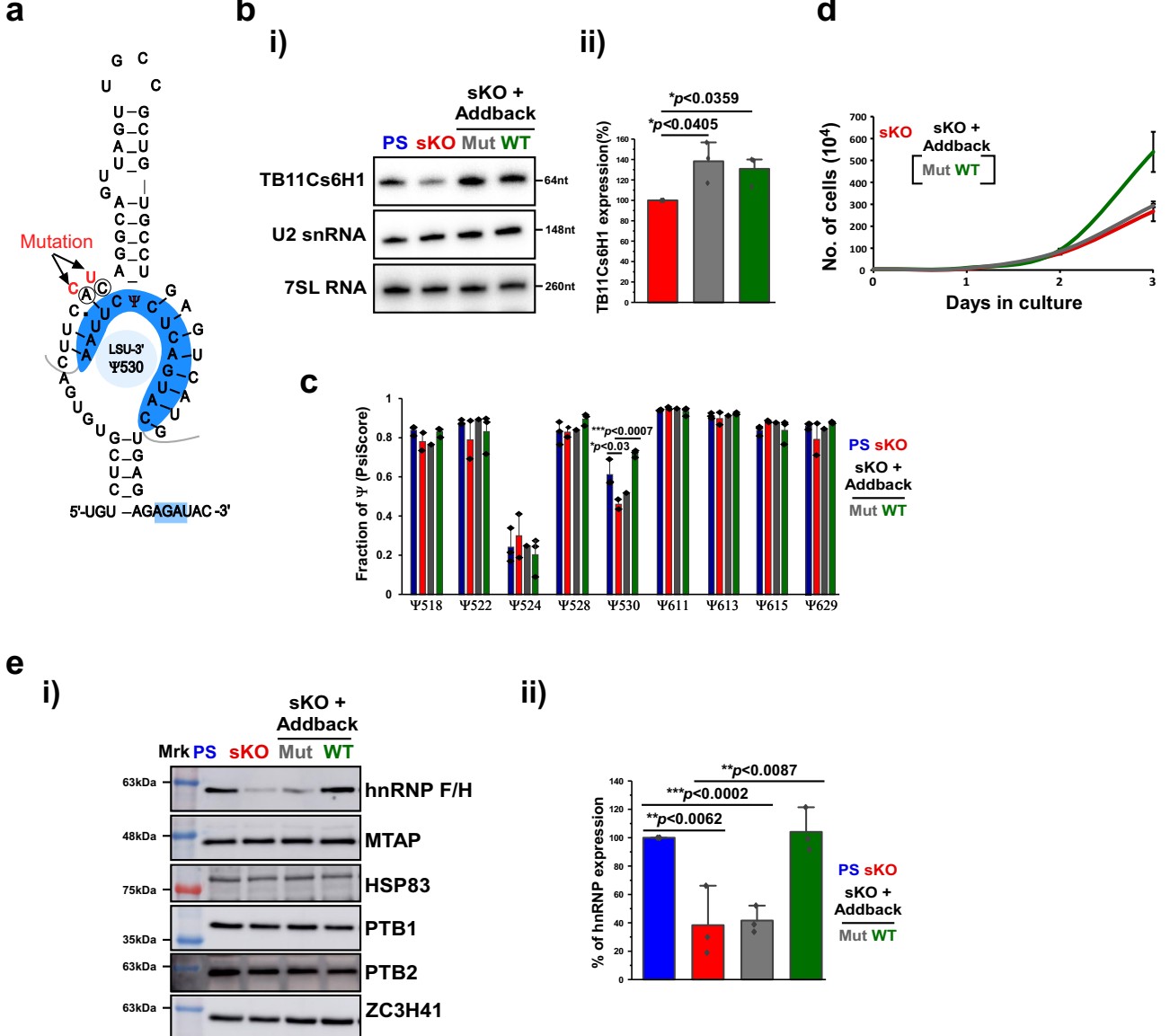

**Fig. 3 | Ψ530 on *T. brucei* LSU-rRNA regulates translation. a** Schematic representation indicating potential base pairing of TB11Cs6H1 with the rRNA to guide Ψ530 and the mutated form with two mismatches introduced. **b** (i and ii) Validation of TB11Cs6H1 snoRNA addback in sKO cells by Northern analysis. 7SL and U2 RNA served as loading control. Data are presented as mean ± S.D. Experiments were done in triplicate (*n* = 3). *p*-value was determined by Student's *t*-test one-tailed distribution. The exact *p*-values are presented within Figure. **c** Wild-type TB11Cs6H1 addback recovers Ψ 530 upon sKO. The PsiScore data are shown as mean ± S.D. Three independent replicates of wild-type TB11Cs6H1 (WT), and one replicate of mutated-TB11Cs6H1 (Mut) were used. *p*-value was determined by Student's *t*-test Two-tailed distribution. The exact *p*-values are presented within Figure. **d** Growth of cells upon TB11Cs6H1 addback in sKO cells. The growth of TB11Cs6H1 sKO, addback of WT-TB11Cs6H1, and mutant-TB11Cs6H1 were compared at 27 °C during the time period indicated. Data are presented as mean ± S.E.M. Experiments were done in triplicate (*n* = 3). Each cell line was grown in three independent cultures,

and growth was monitored simultaneously. **e** Translation of hnRNP F/H following snoRNA addback. (i) Whole cell lysates from PS, clonal population of TB11Cs6H1 sKO and the addbacks of TB11Cs6H1 were subjected to Western analysis with the indicated antibodies. The dilutions used for the antibodies were: hnRNPF/H (1:1,000), MTAP (1:10,000), PTB1 (1:10,000), PTB2 (1:10,000), and ZC3H41 (1:10,000). ZC3H41, PTB1, PTB2, MTAP and HSP83 serve as loading controls. Same volume of sample was loaded in different SDS-PAGE gels and western analysis was performed in parallel. At least one blot was used to obtain loading control from the same blot. (ii) Relative expression of hnRNP F/H in addback cells. Data are presented as mean ± S.D. Experiments were done in triplicate (*n* = 3). *p*-value was determined by Student's *t*-test one-tailed distribution. No adjustments were made for multiple comparisons. The exact *p*-values are presented within Figure. All samples shown in Figures bi and ei were derived from the same experiment and blots were processed in parallel. Source data are provided as a Source Data file.

Interestingly, *T. brucei* ribosomes have only two highly conserved m5C (LSUβ_ m5C542, LSUβ_ m5C1324) and lack a homolog of *Leishmania* SSU_m5C2061. Tandem LC-MS analysis of base-modifications in PCF and BSF rRNA suggests that all these sites are fully modified in both life stages (Supplementary Data 11).

Closer inspection of both PS and sKO models suggested that ribosomes lacking Ψ530 (sKO) have poor and hardly any EM density in the position of eS12, unlike PS ribosomes wherein the entire protein

could be modeled (Fig. 5c). One possible explanation for this phenomenon is that most of the sKO ribosomes lack eS12, while only a small subset of ribosomes might contain eS12, as seen in the MS data. Note that we verified the presence of all other RPs in both structures (Supplementary Data 9). To visualize the structural differences between PS and ribosomes lacking Ψ530 (sKO), we superimposed the two structures and observed very similar global architecture. Minor structural variations were observed in regions distal from the H69,

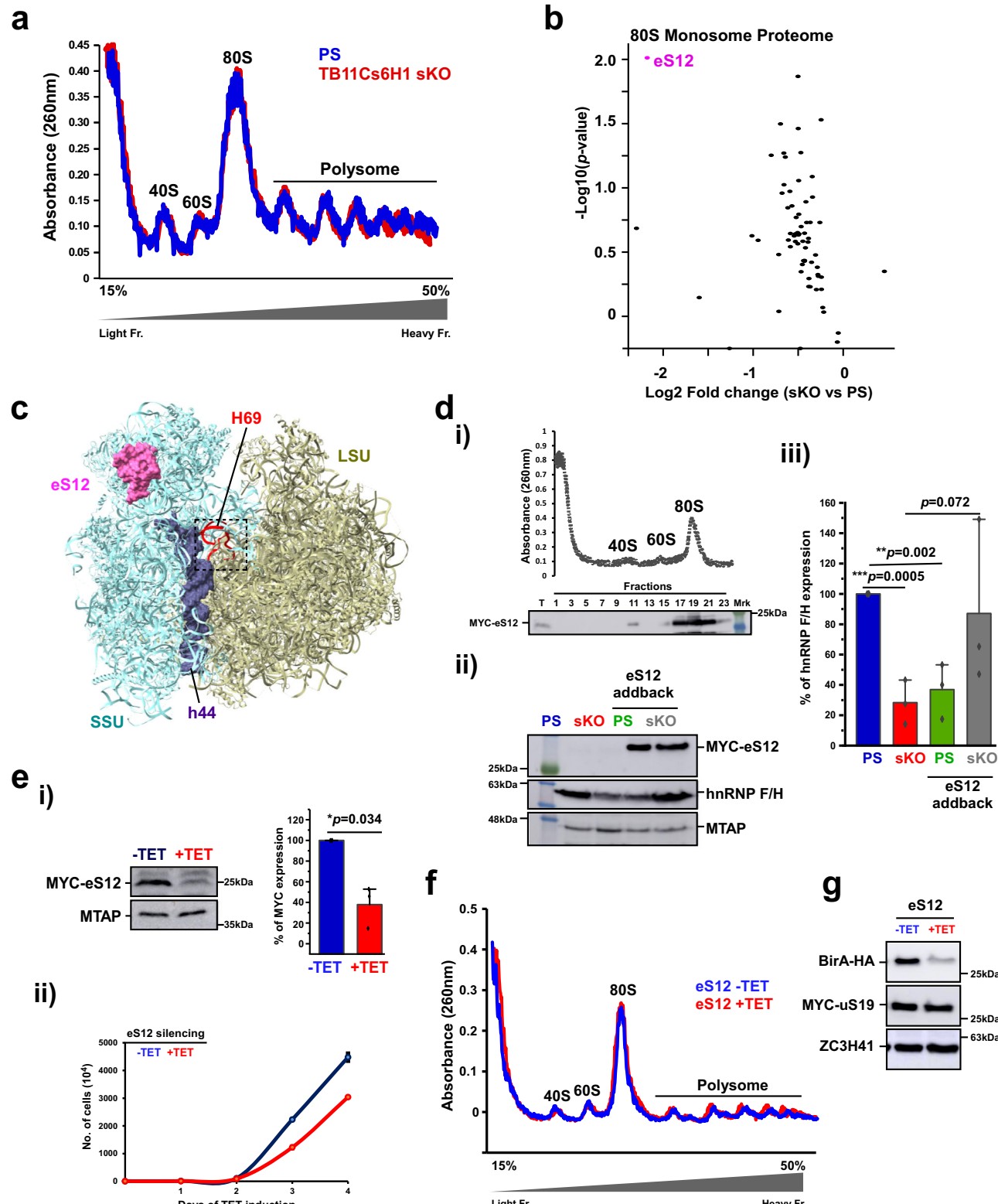

which may result in the reduction of ribosome activity (Supplementary Fig. 10). This is the first structural evidence demonstrating that a lack of a specific RP in mature ribosomes results from the loss of a single Ψ.

## Discussion

In recent years it became evident that ribosomes do not exist as a homogenous population in the cell, and that ribosomes with specialized function can differ in their RP identity or composition[34,42]. Here,

we provide evidence that links the change of a single Ψ located in H69 on the rRNA, with a change in 80S monosome composition, resulting in structural changes bearing functional implications on the ribosome.

Using HydraPsiSeq, changes were found in Ψ modification in several types of cancer cells, and during chondrogenic differentiation, but these changes have yet to be correlated with any effect on translation[18]. Recently, it was demonstrated that eliminating a single H/ACA snoRNA in human cells guiding two Ψs on rRNA, did not change

**Fig. 4 | TB11Cs6H1 sKO affects 80 S monosome composition. a** Fractionation of ribosomes by sucrose gradient. The fractions containing 40 S, 60 S, and 80 S monosomes, and polysomes are indicated. **b** Composition of 80 S monosomes. Three biological replicates were used to calculate the fold-change (FC). Data are presented as log2FC in the *x*-axis and log10 (*p*-value) along the *y*-axis. Significant changes in protein abundance are indicated by pink dots, and non-significant proteome changes are shown in black. *p*-value was determined by one-sample t-test. Benjamini-Hochberg correction for multiple hypothesis testing (SignificanceB) was performed. **c** Localization of eS12 on the trypanosome ribosome. The identity and localization of eS12, h44, and the H69 domain are indicated. **d** eS12 addback in sKO cells recovers hnRNPF/H translation. (i) Localization of MYC-eS12 on ribosomes. Whole cell extract from sKO cells expressing MYC-eS12 were fractionated on a 10–30% sucrose gradient. The indicated fractions were precipitated and subjected to western analysis using MYC antibody (9E10). (ii) Western analysis with the indicated antibodies. (iii) hnRNP F/H expression. Data are presented as mean ± S.D. Experiments were done in triplicate (*n* = 3). *p*-value was determined by Student's *t*-test one-tailed distribution. The exact *p*-values are presented within Figure. **e** eS12 silencing. (i) Western analysis indicating efficient silencing of eS12. MTAP served as loading control. Data are presented as mean ± S.D. Experiments were done in triplicate (*n* = 3). *p*-value was determined by Student's *t*-test Two-tailed distribution. *p = 0.034. (ii) Growth of cells following *eS12* silencing. Data are presented as mean ± S.E.M. Experiments were done in technical triplicates (*n* = 3). **f** Fractionation of ribosomes derived from *eS12* silenced cells on sucrose gradient. Absorbance at 260 nm was measured. **g** Aldolase translation upon eS12 silencing is dependent on its 3′ UTR. Whole cell lysates from cells induced (+TET) and uninduced (−TET) for eS12 silencing were subjected to western analysis with the indicated antibodies. MYC-uS19 served as the internal control, and ZC3H1 served as a loading control. All samples shown in Figures dii and g were derived from the same experiment and blots were processed in parallel. Source data are provided as a Source Data file.

80S and polysomal properties, but increased translational miscoding and stop codon readthrough frequencies[47]. The current study demonstrates a profound effect on ribosome function due to a change in a single Ψ modification on rRNA.

In this study, we re-visited the mapping of the Ψ on rRNA, focusing on the level of modification at each site using four approaches, Ψ-seq, HydraPsiSeq, nanopore RNA sequencing and tandem LC-MS. The results confirmed the existence of 70 Ψs on rRNA that are guided by snoRNAs. All these methods indicated the presence of hyper and hypo modified positions on trypanosome ribosomes. Interestingly, the variable positions are in the decoding center, around the PTC, and the polypeptide entrance tunnel. In addition, we observed that at least 38 Ψ sites in the polysomal RNA (36 sites in total RNA) are almost fully modified in the two life stages, whereas other sites are only partially modified. The slight discrepancy between the mapping methods may be attributed to factors such as the sensitivity of reverse transcriptase used in Ψ-seq to the secondary structure of the RNA. HydraPsiSeq depends on differential chemical cleavage and LC-MS on enzymatic digestion. These methods could be less efficient for quantifying neighboring sites, such as those present in H69. Our study compares all four available genome-wide mapping methods to map Ψ on rRNA from the same source. More such studies are needed to fully understand the source of differences observed, and therefore, differentially modified sites should be identified by at least two to three different methods.

In mammals ~20% of Ψs on rRNA are altered in cancer cells or during development of stem cells[18]. In the current study, we demonstrate that ~50% of the Ψ sites are partially modified and that the Ψs near the PTC and decoding center are differentially modified between the two life stages of the parasite (Fig. 1b). Here, we also report differences that exist in Ψ level between total and polysomal rRNA, which may suggest that only ribosomes with a particular pattern of Ψ are capable of becoming translating ribosomes. Differentially modified Nm positions in trypanosomes were shown to be clustered around functional domains such as the A and P sites in the SSU and the PTC[20].

Functional studies on H/ACA snoRNAs in trypanosomes are limited due to their genomic organization, as these RNAs are encoded by reiterated gene clusters carrying a mixture of C/D and H/ACA snoRNAs[32,33]. snoRNAi is less efficient for H/ACA RNAs compared to C/D snoRNAs due to the presence of long double-stranded RNA domains[48]. We therefore could not study the role of snoRNA in guiding differentially modified Ψs on ribosomes. In contrast, the snoRNAs guiding on H69 are encoded by a solitary gene carrying a single snoRNA and are amenable to CRISPR-Cas9 manipulation. In addition, the biological significance of these Ψ modifications on H69 was previously studied in both prokaryotes and eukaryotes, and they were shown to impact translation only when multiple Ψs were simultaneously disrupted. Note that based on HydraPsiSeq, Ψ530 was shown to differ in its level between polysomal and total RNA. Indeed, our study supported the choice for functional studies on Ψ530, since a clear phenotype was observed upon its depletion.

Our previous study suggested differential levels of Ψs on H69 between the life stages of the parasites based on Ψ-seq analysis[31]. However, this notion was not supported by the additional quantitative methods. Thus, our current study demonstrates how important it is to combine different methods to reach a consensus regarding the level of modification on individual sites. Despite the discrepancy described here regarding Ψ levels in H69, our results highlight the importance of Ψ530 for ribosome structure and function even beyond that described in other systems, such as bacteria and yeast[22,24,25].

Several studies related changes in RP composition with specific effects on translation[34,42]. The changes in RP observed here relate mostly to eS12. Interestingly, the level of eS12 was increased by over 3.4-fold in BSF compared to PCF ribosomes (Supplementary Fig. 11). Our study based on RNAi suggests that as opposed to other eukaryotes, the depletion of eS12 did not affect polysome formation, explaining why the sKO cells of TB11Cs6H1 are viable. Moreover, our cryo-EM data demonstrate that ribosomes from the TB11Cs6H1 sKO that were visualized lacked eS12.

The finding that a single modification can affect the rRNA structure in domains that are distal from the site of the altered modification, is consistent with a study in yeast demonstrating that elimination of $m^5C$ changed the structure of the RNA leading to loss of multiple ribosomal proteins[49]. In addition, changes in ribosomal proteins L11 (uL5) and S18 (uS13), which form a dynamic inter-subunit interaction called the B1b/c, had effects on cellular viability and translational fidelity. Despite the localization of the mutations to L11 (uL5), changes were revealed in rRNA structure in both the large and small subunits. These results suggest that, while spatially remote, all these different regions of the ribosome are inter-connected[50].

The current study highlights the importance of Ψ on H69 for ribosome function. It also indicates that not all Ψ have the same effect on the ribosome function, for example, the phenotype of Ψ530 alterations were different from those of Ψ522. H69 interacts with h44 of SSU as well as with the initiation factor eIF5A[51]. Studies using NMR structure *of E.coli* H69 domain suggested that the presence of Ψ enables base-stacking, supporting local conformational dynamics, which are important for the formation of the intersubunit bridge during the translational process[24]. These important H69 functions explain why a single change in its modification has such a profound effect on the ribosome structure and its function.

The current study demonstrates that the effect of snoRNA sKO on the translation of a subset of proteins does not result from differential codon usage, but depends on the mRNA 3′ UTR, for at least two substrates studied. It is currently unknown how the 3′ UTR dictates this regulation. It is possible that the differential translation depends on

**Table 1 | Cryo-EM data-collection, model refinement and validation statistics**

| 80 S Ribosome | Parental strain (EMD-17208, PDB-8OVA) | TB11Cs6H1 sKO (EMD-17212, PDB-8OVE) |
|---|---|---|
| Data Collection | | |
| Microscope | Titan Krios | Titan Krios |
| Camera | Gatan K3 | Gatan K3 |
| Voltage (kV) | 300 | 300 |
| Magnification | 105000 | 105000 |
| Pixel size (Å.px$^{-1}$) | 0.85 | 0.85 |
| Defocus range (μM) | (−0.5)–(−1.50) | (−0.5)–(−1.50) |
| Total dose (e/Å$^2$) | 53.8 | 60 |
| Micrographs collected | 6069 | 5530 |
| Refinement | | |
| Number of particles (autopicked) | 765,885 | 883,803 |
| Number of particles (used for 3D reconstruction) | 459,985 | 552,813 |
| Resolution (Å; at FSC = 0.143) | 2.47 | 2.6 |
| CC (model to map fit) | 0.91 | 0.83 |
| Model Quality | | |
| Bonds (Å) | 0.004 | 0.004 |
| Angles (°) | 0.612 | 0.640 |
| Chirality (°) | 0.038 | 0.039 |
| Planarity (°) | 0.004 | 0.005 |
| Validation | | |
| Clashscore | 4.50 | 5.84 |
| Proteins | | |
| MolProbity score | 1.92 | 2.22 |
| Rotamer outliers (%) | 3.72 | 5.39 |
| Ramachandran favored (%) | 96.18 | 95.01 |
| Ramachandran allowed (%) | 3.76 | 4.93 |
| Ramachandran outliers (%) | 0.05 | 0.06 |
| RNA | | |
| Correct sugar pucker (%) | 99.38 | 99.40 |
| Correct backbone conformation | 80.42 | 79.41 |

RNA binding protein(s) that differentially regulate translation in the sKO ribosome lacking eS12. Note that eS12 is located in the head region of the small subunit ribosome, and its absence can impact the pre-initiation complex and its interaction with protein(s) bound to mRNA 3′ UTR.

Trypanosomes LSU rRNAs are unique compared to those of other eukaryotes, since they are fragmented and composed of two LSU chains and four small rRNA fragments[52]. Such a fragmented ribosome might be more fragile than ribosomes harboring a continuous LSU rRNA molecule, hence even a single change in modification might severely impact translation and cell growth, as observed in this study. rRNA modification is also essential for adaptation of the closely related human parasite, *Leishmania donovani*[53]. Our study highlights the power of rRNA modification, which can be harnessed for developing new RNA therapeutics against these parasites and other diseases associated with alteration in rRNA modification. The change of rRNA modification in these parasites during cycling between the two hosts or upon fitness gain provides proof of concept for such an approach[31,53].

## Methods

### Cell growth and transfections

Procyclic form (PCF) *T. brucei*, strain 29-13[54], which carries integrated genes for the T7 polymerase and the tetracycline repressor, was grown in SDM-79 medium supplemented with 10% fetal calf serum, in the presence of 50 μg/ml hygromycin. Cells were grown in the presence of 15 μg/ml G418 for generating the RNAi silenced cell lines. PCF 1313 strain (a gift from C. Clayton, ZMBH, Heidelberg, Germany) which carries the tetracycline repressor gene used for expressing the Cas9 protein was grown in SDM-79 medium supplemented with 10% fetal calf serum, in the presence of phleomycin. The bloodstream form (BSF) of *T. brucei* 427 (cell line 1313-514) was aerobically cultivated at 37 °C under 5% $CO_2$ in HMI-9 medium supplemented with 10% fetal calf serum, 2 μg/ml G418, and 2.4 μg/ml phleomycin[55].

### snoRNA CRISPR-Cas9

For the expression of spCas9 in *T. brucei*, a plasmid (Cas9-NLS-PPOTv4) was generated by cloning *Streptococcus pyogenes* Cas9 from plasmid pTB011[38] into PPOTv4[37] (a gift from Dr Samuel Dean, University of Oxford, UK) between the *HindIII* and *BamHI* restriction sites. This plasmid was linearized using *BsaBI* to enable integration into the PFR locus, transfected into procyclic 1313 *T. brucei* cells (a gift from Prof. Christine Clayton, ZMBH, University of Heidelberg, Germany) and cloned. snoRNA-specific guide RNAs (gRNAs) were designed using the EuPaGDT database (http://grna.ctegd.uga.edu/) and in vitro transcribed. To prepare homologous DNA repair (HDR) templates, PPOTv4 plasmid was used as a template for PCR amplification; it carries coding sequences for eYFP/tagRFPt and neomycin/hygromycin resistance. The primers used for gRNA synthesis and HDR template are listed in Supplementary Data 12. Both gRNAs and the snoRNA-specific PCR product were transfected, selected using antibiotics, single cells were cloned using Fluorescence-activated cell sorting (FACS), and KO cells were screened by Northern analysis and PCR.

snoRNA including its flanking sequence was cloned into pTB011 (a gift from Dr. Eva Gluenz, Institute of Cell Biology, University of Bern, Switzerland) between the *HindIII* and *BamHI* restriction site to generate a snoRNA overexpression plasmid. Site-directed mutagenesis was performed using the primers listed in Supplementary Data 12 to generate mutations in the snoRNA. The template plasmid was digested with *DpnI*, and clones were isolated and verified by Sanger-sequencing (Macrogen Inc). This plasmid was linearized using *PacI* to enable integration into the tubulin locus, transfected and cloned.

### Northern analysis

For Northern analysis, total RNA was extracted and separated on a 10% acrylamide denaturing gel. Antisense RNA probes were prepared by in-vitro transcription using α-$^{32}$P-UTP[32,56]. Primers used for in-vitro transcription are listed in Supplementary Data 12.

### Hydrazine and aniline treatment

Total RNA and polysomal RNA (5μg) was treated with 50% hydrazine (Sigma) for 45 min on ice, and ethanol precipitated[18]. The RNA pellet was then resuspended in 1 M aniline (Sigma) (pH 4.5 adjusted using glacial acetic acid) until the white pellet was completely dissolved, boiled for 15 min at 60 °C in the dark, and immediately placed on ice. The fragmented RNA was recovered by ethanol precipitation and used for library preparation[20].

### HydraPsiSeq library preparation

To perform HydraPsiSeq, the fragmented RNA (~800 ng) was dephosphorylated with FastAP Thermosensitive Alkaline Phosphatase (Thermo Scientific), cleaned by Agencourt RNA clean XP beads (Beckman Coulter) and ligated to a 3′ linker using high concentration T4 RNA Ligase 1 (NEB) in a buffer containing DMSO, ATP, PEG 8000, and RNase inhibitor (NEB). The ligated RNA was cleaned from excess

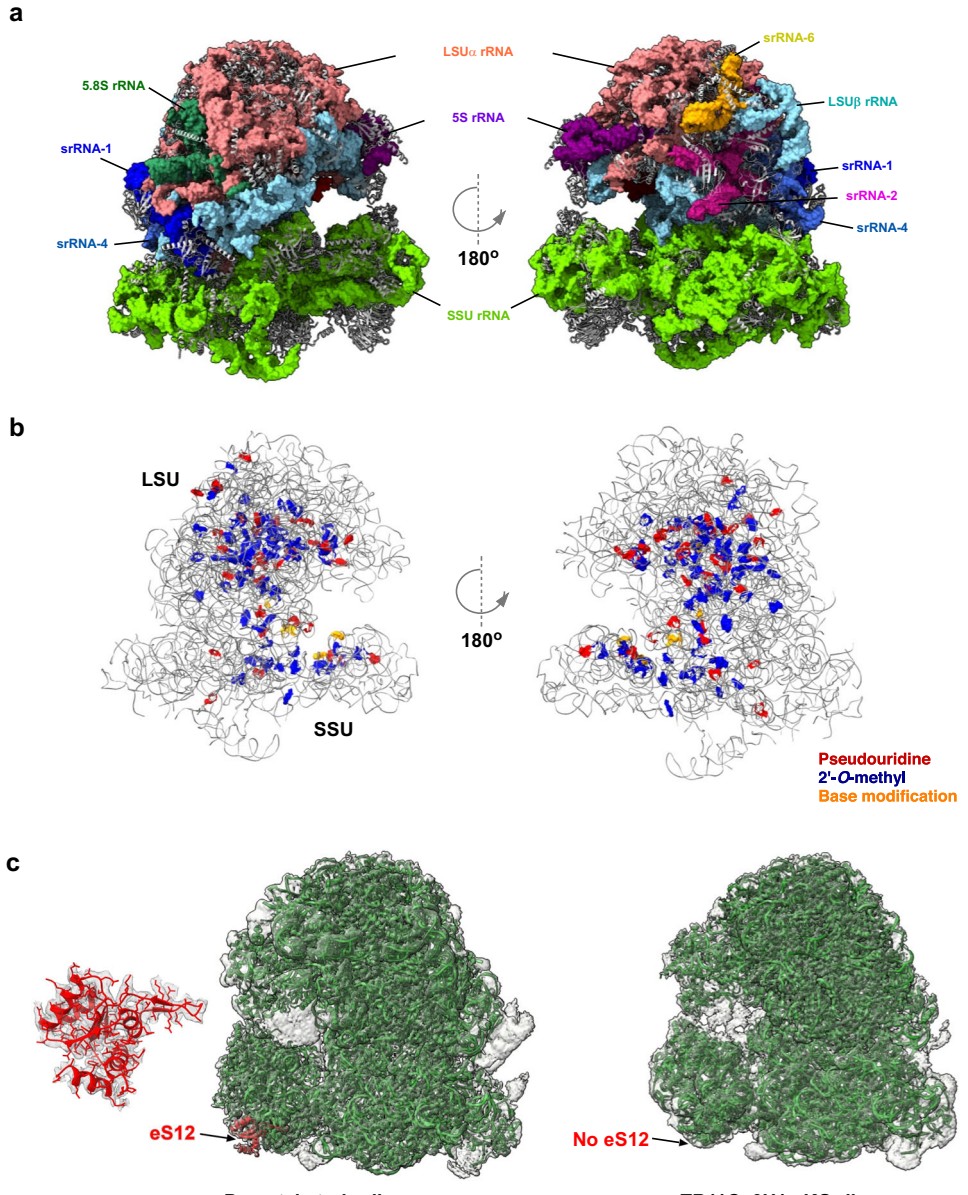

**Fig. 5 | High-resolution cryo-EM structure of *T. brucei* 80S ribosome at 2.47 Å. a** The atomic structure of *T. brucei* ribosome is shown in both front and back views. The different RNA chains are shown in a colored space filled presentation, and the various RP chains are shown as gray ribbons. **b** Location of rRNA modifications in the *T. brucei* ribosome. 2'-*O*-methylation, Ψ, and base-modifications are presented in red, blue, and orange, respectively. **c** EM density around eS12. The EM map and model of PS and sKO ribosomes are superimposed. The location of eS12 is indicated, and the modelled eS12 in PS ribosome is shown in red. The contour level of both maps is 2 σ. Representative micrographs of cryo-EM data showing individual particles are provided in source data. Source data are provided as a Source Data file.

linker using Dynabeads® MyOne™ SILANE beads (Thermo Scientific), and first strand cDNA was prepared using the AffinityScript Reverse Transcriptase enzyme (Agilent). The RNA was subsequently degraded using 2 µl of 1 M NaOH, and the cDNA was cleaned using Dynabeads® MyOne™ SILANE beads (Thermo Scientific). The cDNA was further ligated to a 3' adapter using high concentration T4 RNA Ligase 1 (NEB) and cleaned of excess adapter by using Dynabeads® MyOne™ SILANE beads (Thermo Scientific). The adapter ligated cDNA was PCR enriched using NEBNext® High-Fidelity (NEB) polymerase (9 PCR cycles), separated on an E-Gel EX agarose gel (Invitrogen) and size selected in the range of 150–300 bp (containing ~30–180 nt corresponding to RNA). The amplicons were gel purified using NucleoSpin Gel and PCR Clean-up kit (Macherey-Nagel), and sequenced in a Nextseq system (Illumina) in paired end mode (20 million reads for each sample).

## HydraPsiSeq data analysis

The paired end reads obtained from each sample were aligned to the *Trypanosoma brucei* ribosomal RNA using Smalt v_0.7.5 (http://www.sanger.ac.uk/resources/software/SMALT/) with default parameters. For each sample, the resulting bam file was sorted and filtered for proper pairs using SAMtools v1.9[57] and then converted to a BED file using the bamtobed module from the BEDtools v2.26.0 Suite[58]. Using an in-house Perl script on each bed file, the number of reads whose 5'-end alignments initiate at that base for each position on the rRNA was calculated. The total coverage for each base was calculated using genomecov module from the BEDtools v2.26.0 Suite[58]. These files were then used as input for the R scripts (https://github.com/FlorianPichot/HydraPsiSeqPipeline), as previously described[18].

### Nanopore direct RNA long read sequencing

*T. brucei* rRNA derived from total RNA was prepared for long-reads direct RNA sequencing as described before[19] with minor modifications. In brief, the total RNA was treated with T4 PNK without ATP followed by RNA cleaning and size selection of molecules larger than ~200 bases (ZIMO RNA Clean & Concentrator-5, as per manufacture's protocol for selection of large RNA). ~15 µg of the size selected RNA was later treated with poly-A polymerase (NEB M0276S) for 30 min at 37 °C as recommended by manufacturer's protocol (100 µl protocol), and then poly-A + RNA was purified by NEB poly-A beads. The presence of rRNA after poly-A selection was validated using TapeStation High Sensitivity RNA ScreenTape analysis (Aglient). Barcoding and library preparation was performed as described by the DeePlexiCon protocol[59]. From each sample (PCF, BSF and *CBF5* silenced rRNA), ~50 ng was run on a MinION device using flowcells R.9.4.1 and the direct RNA long read sequencing kit.

### Demultiplexing direct RNA sequencing, basecalling and alignment

Demultiplexing of the barcoded direct RNA sequencing libraries was performed using DeePlexiCon with stringent criteria of 0.9[59]. Reads were base-called with guppy_basecaller using the following parameters: -r -c rna_r9.4.1_70bps.cfg. Reads were then mapped to the *T. brucei* reference genome by minimap2 (-ax map-ont)[60] and transformed to BAM files, sorted and indexed using SAMtools[57].

### Detection and quantification of Ψ sites by nanopore sequencing

To detect putative Ψ sites by nanopore sequencing the following steps were performed. Initially, for each condition (PCR, BSF and *CBF5* silenced), the mismatch ratio of T to C was calculated for each T site in the rRNA. The T to C mismatch of Ψ sites was compared to T-non-Ψ sites to assess the mismatch ratio upon pseudouridylation. To validate the performance of Ψ detection by this method, a ROC analysis was performed using the Ψ sites detected by Ψ-seq and HydraPsiSeq as true-positive sites. To determine the changes in modification level, the T to C mismatch delta of each position was calculated, i.e., (T to C mismatch of PCF) - (T to C mismatch of CBF5 silenced). The R package 'cutpointr' was used for ROC analysis. 'cutpointr' command with the parameters of: method = maximize_metric, metric = F1_score was used to calculate the optimal 'cutpoint' of the delta T to C mismatch, for AUC value and for confusion matrix. 'plot_roc' command was used to plot ROC curve.

### Poly-A RNA-seq analysis

The RNA-sequencing reads were aligned to the *T. brucei* genome (v5) using Smalt v_0.7.5 (http://www.sanger.ac.uk/resources/software/SMALT/) with default parameters. After alignment, the expression level of mRNA was quantified using HTSeq-count[61] to count the number of reads aligned to each gene in the TriTrypDB *T. brucei* gtf file (https://tritrypdb.org). The gene count tables were used as input for the R-Bioconductor package DESeq2 (v1.38.3)[62] to perform differential gene expression analysis using the default parameters. mRNAs with an absolute log2 fold-change of >/=1 and FDR < 0.05 were considered differentially expressed.

### Purification of rRNA subunits for Tandem LC-MS

The rRNAs of *T. brucei* were purified from the total RNA by using reversed-phase LC through a PLRP-S 4000 Å column (4.6 × 150 mm, 8 µm, Agilent Technologies). The rRNAs were eluted with a 120-minute gradient of 10.8–13.6% (10.8–12.4% for the first 20 min and 12.4–13.6% for the next 100 min) (v/v) acetonitrile in 100 mM TEAA (pH 7.0) and 0.1 mM diammonium phosphate from the column, at a flow rate of 200 µl/min at 60 °C while monitoring the eluate at A260[63].

### LC-MS and MS/MS analysis, and database search of RNA fragments

Purified rRNA was digested with RNase T1 or A, and the resulting RNA fragments were cyanoethylated for labeling pseudouridine[10]. The RNA fragments were analyzed with a direct nanoflow LC-MS system, as described[64]. The column was prepared with a fused-silica capillary (150-µm i.d. x 240 mm in length) packed with a reversed-phase material (Develosil C30-UG-3, 3-µm particle size; Nomura Chemical). The LC was performed at a flow rate of 200 nl/min using a 120-min linear gradient from 10% methanol to 7.6% methanol/9.8% acetonitrile in 10 mM triethylammonium acetate (pH 7.0)[35]. The eluate was sprayed online at −1.3 kV with the aid of a spray-assisting device to a Q Exactive Plus mass spectrometer (Thermo Fisher Scientific) in negative ion mode[35]. Ariadne was used for database searches and assignment of MS/MS RNA spectra[65]. The composite of Trypanosoma brucei rRNA sequences was used as a database. The following default search parameters for Ariadne were used: maximum number of missed cleavages, 1; variable modification parameters, two modifications per RNA fragment, including cyanoethylation for uridine and methylation for any residue; RNA mass tolerance, ±5 ppm, and MS/MS tolerance, ±20 ppm.

### Proteolysis and dimethyl labeling

Cells lysed using RIPA (Sigma) buffer were precipitated in 80% ethanol, and the protein pellets were dissolved in [8.5 M Urea, 10 mM DTT, 400 mM ammonium bicarbonate (ABC)], sonicated twice (90%, 10-10 cycles, 5'), and centrifuged. The proteins in the solution were reduced (60 °C for 30 min), modified with 35.2 mM iodoacetamide in 100 mM ammonium bicarbonate (in dark, at room temperature for 30 min) and digested in 1.5 M Urea, 70 mM ABC with modified trypsin (Promega) at a 1:50 enzyme-to-substrate ratio, overnight at 37 °C. A second digestion was done for 4 h. The resulting peptides were desalted using C18 Top tip columns (Glygen), dried, and re-suspended in 50 mM Hepes (pH 7.9). Labeling by dimethylation was done in the presence of 100 mM NaCBH₃ (Sterogene, 1 M), by adding Light Formaldehyde (35% Frutarom, 12.3 M) to one of the samples, and "Heavy" Formaldehyde (20% w/w, Cambridge Isotope laboratories,6.5 M) to the other sample to a final concentration of 200 mM. The samples were incubated for 1 h. Samples were neutralized with 1 M ABC for 30 min, and equal amounts of the light and heavy peptides were mixed, cleaned on C18 Top Tip columns, and re-suspended in 0.1% formic acid.

### Mass spectrometry analysis of proteins

The peptides were resolved by reversed-phase chromatography on 0.075 × 180-mm fused silica capillaries (J&W) packed with Reprosil reversed phase material (Dr Maisch GmbH, Germany). The peptides were eluted with a linear 180 min gradient of 5 to 28%; a 15 min gradient of 28 to 95%; and 25 min with 95% acetonitrile and 0.1% formic acid in water, at a flow rate of 0.15 µl/min. Mass spectrometry was performed by Q-Exactive HFX mass spectrometer (Thermo) in positive mode using repetitively full MS scan followed by collision to induce dissociation (HCD) of the 30 most dominant ions selected from the first MS scan. The mass spectrometry data were analyzed using the Proteome Discoverer 1.4 (Thermo) software, searching against the *T. brucei* from the TriTrypDB v.35 database (https://tritrypdb.org). Results were filtered with rank 1 peptides and 1% false discovery rate. The ratios were normalized according to the protein's median ratio. Perseus software (https://maxquant.net/perseus/) was used for statistical analysis of the data. Significant outliers relative to a given population were calculated using intensity-dependent calculation. The truncation was based on the Benjamini-Hochberg correction for multiple hypothesis testing (SignificanceB)[66]. For combined analysis of different replicates, one-sample t-test was used to determine if the mean was significantly different from a fixed value (0). All MS analyses were performed at the Smoler Proteomics Center, Technion, Israel and raw data are available via ProteomeXchange, with identifier

PXD030544. The identity of heavy and light isotope labelled samples are listed in Supplementary Data 13.

## Dicistronic reporter cassette and in situ tagging

To create a dicistronic reporter cassette, the BirA-HA gene (a gift from Prof. Paul A Khavari, Stanford University School of Medicine, USA) was cloned between the *NheI* and *XhoI* restriction sites of the PPOTv4 vector (a gift from Dr. Samuel Dean, Sir William Dunn School of Pathology, University of Oxford, UK) carrying the aldolase 3′ UTR. This vector was further used to clone MYC-uS19 replacing the eYFP gene between the *HindIII* and *BamHI* restriction sites, and short hnRNP F/H 5′ UTR between *MluI* and *HindIII* restriction sites. The resulting construct was linearized with the *BsaBI* enzyme for integration into the PFR locus. PS and TB11Cs6H1 cells were transfected with the resulting vector, and clones were used for western analysis. MYC-uS19 served as an internal control.

C-terminal MYC tagged hnRNP F/H constructs were prepared by cloning ~500 nt of the hnRNP F/H coding sequences into pNAT X-tag (a gift from Dr. Sam Alsford, London School of Hygiene & Tropical Medicine, UK) between the *HindIII* and *XbaI* restriction sites. The primers used are listed in Supplementary Data 12. The plasmids were linearized with *NruI* for integration into the authentic chromosomal loci.

## Western analysis

Whole-cell lysates ($10^7$ cells) were fractionated by SDS-PAGE, transferred to Protran membranes (Whatman), and reacted with the indicated antibodies. The bound antibodies were detected with goat anti-rabbit immunoglobulin G coupled to horseradish peroxidase and were visualized by ECL enhanced chemiluminescence (Amersham Biosciences). The dilutions used for the antibodies were: HA (1:1,000), MYC (1:1,000), MTAP (1:10,000), and ZC3H41 (1:10,000), hnRNPF/H (1:1,000), PTB1 (1:10,000), PTB2 (1:10,000), U2AF35 (1:10,000), HSP83 (1:10,000), RBP6 (1:1000) and RBP10 (1:1,000). HA (BioLegend 16B12) and MYC (Santa-cruz 9E10) antibodies were purchased commercially, whereas other antibodies were prepared in our lab previously or obtained as a gift from Prof. Christian Tschudi (Yale University) and Christine Clayton (University of Heidelberg).

## Fractionation of ribosomes on sucrose gradient

Whole cell extracts were prepared from $5 \times 10^9$ *T. brucei* PCF and BSF cells in a buffer containing 150 mM KCL, 20 mM Tris-pH 7.6, 10 mM MgCl$_2$, 0.5 M DTT and 0.1% NP-40. In addition, 1 µl leupeptin (10 mg/ml) protease inhibitor, and 1 µl of RNasin (Thermo Scientific) were added to the lysate. Ribosomes were fractionated on a 10–30% or 15–50% (w/v) sucrose gradient by centrifugation for 3 h at 210,000 g in a Beckman SW41 rotor at 4 °C. The fraction carrying ribosomes was ethanol precipitated, and ribosomes were resuspended in buffer-150 (150 mM KCL, 20 mM Tris-pH 7.6, 10 mM MgCl$_2$) and used for mass-spectrometry or RNA analysis.

## Methionine incorporation assay

Procyclic cells ($5 \times 10^7$) were washed in methionine free (phenol-red free) SDM-79 media, treated with 50 µM Click-iT® AHA (L-azidohomoalanine) reagent (Invitrogen) in methionine-free (phenol-red free) SDM-79 media, and incubated for 30 min at 27 °C according to the manufacturer's protocol. The cells were fixed using 1.6% formaldehyde, and permeabilized with Triton:Tween 20 (1:0.1%) in 1xPBS. The cells were incubated with Click-iT® reaction cocktail for 30 min and washed once with 3% BSA in 1xPBS. The incorporated Click-iT® AHA was determined by FACS Aria using the Alexa 488 channel.

## Purification of 80 S ribosomes for cryo-EM

Mid-log phase *T. brucei* cells (~1 g pellet) were washed three times in resuspension buffer (45 mM HEPES-KOH pH 7.6, 100 mM KOAc, 10 mM Mg(OAc)$_2$ and 250 mM Sucrose) and suspended in cold buffer-A (45 mM HEPES-KOH pH 7.6, 100 mM K(OAc), 10 mM Mg(OAc)$_2$, 250 mM Sucrose, 5 mM β-mercaptoethanol and a 1:40 dilution of RNasin U (Promega). Cells were homogenized using 0.01% NP40, and cell debris were removed by centrifugation (30 min at 21,000 × *g*) at 4 °C. The lysate was gently loaded onto a 1.1 M sucrose cushion in cold buffer-B (20 mM HEPES-KOH pH 7.6, 150 mM KOAc, 10 mM Mg(OAc)$_2$, 1.1 M sucrose and 5 mM β-mercaptoethanol) and centrifuged at 310,000 × *g* at 4 °C for 15 h (Ti70 rotor, Beckman). Following centrifugation, the pellet was resuspended in cold buffer-C (20 mM HEPES-KOH pH 7.6, 150 mM KOAc, 10 mM Mg(OAc)$_2$ and 5 mM β-mercaptoethanol) and loaded onto a 15–30% (w/v) sucrose gradient in buffer-C (centrifugation at 88,500 × *g*, 11 h, on SW28 rotor, Beckman). The peaks corresponding to 80 S ribosomes were collected, balanced with buffer-C, and centrifuged at 58,500 × *g* for 12 h at 4 °C. The resulting pellet was suspended in buffer-D (20 mM HEPES-KOH pH 7.6, 100 mM KOAc, 10 mM Mg(OAc)$_2$, 10 mM NH$_4$OAc and 1 mM DTT) and centrifuged for 90 min at 245,000 × *g* (TLA-100 rotor, Beckman). The final ribosome pellet was gently resuspended in buffer-D, and aliquots were frozen and stored at −80 °C until further use.

## Cryo-EM data collection and refinement

To prepare cryo-EM grids, 3.5 µl of ribosome suspension (~13 A of A$_{260}$) was applied onto glow-discharged holey carbon grids (Quantifoil R2/2) coated with a continuous thin carbon film. The grids were blotted and plunge-frozen using Vitrobot Mark IV (Thermo Fischer Scientific). A Titan Krios electron microscope (Thermo Fischer Scientific) operating at 300 kV equipped with K3 direct electron detector (Gatan Inc.) was used for collecting cryo-EM micrographs at liquid nitrogen temperature at a nominal magnification of 105,000x, with a pixel size of 0.85 Å/pixel and a dose rate of ~1 electron/Å$^2$/s. Defocus values ranged from −0.5 to −1.5 µm. Relion 3.1 was used for data processing[67]. Motion correction and contrast transfer function parameters were estimated using Motioncor2[68] and CTFFIND-3[69], respectively. The extracted particles were subjected to several rounds of unsupervised 3D classification using a low-pass filtered cryo-EM density map. 3D classes similar to 80 S particles were selected and subjected to auto-refinement in Relion. Following initial refinement, particles were subjected to CTF refinement, Bayesian polishing, and refinement. The resulting high-resolution 3D density map was then subjected to a cycle of multibody refinement using separate masks for the large subunit (LSU), the head, and body regions of SSU[70]. The gold standard Fourier shell correlation (FSC) value criterion of 0.143 was used for determining averaged map resolutions as implemented in Relion 3.1. Local resolution was estimated using Resmap[71].

## Map interpretation: model building and refinement

rRNA and ribosomal protein models were built by template-guided model building in COOT[72]. The coordinates of the *T. brucei* and *L. donovani* ribosome (PDB ID: 4V8M, 6AZ1 and 6AZ3) were used as a template for model building, and were docked onto density maps using UCSF ChimeraX[73]. RNA modifications were manually modelled based on Ψ-seq and HydraPsiSeq, as well as *L. donovani* cryo-EM and mass-spectrometry information. The Mg$^{2+}$, Zn$^{2+}$, Na$^+$, and K$^+$ ion compositions were modeled according to the recently described criteria[74,75]. Model refinement was performed using an iterative approach, including real-space refinement and geometry regularization in COOT[72], followed by real-space refinement using the PHENIX Real_space_ refine tool[76]. The final model was validated using MolProbity[77].

## Statistics and reproducibility

All experiments were repeated three times or more unless indicated. Statistical analysis and bar-graphs were prepared using Microsoft excel, Origin (OriginLab Corporation) and DESeq2 (v1.38.3)[62]. Tandem

**Article** https://doi.org/10.1038/s41467-023-43263-6

LC-MS and Nanopore sequencing analysis of RNA modification and experiments in Figs. 2hii, 2i, 2jii, 4di, and 4g were done only once.

## Reporting summary

Further information on research design is available in the Nature Portfolio Reporting Summary linked to this article.

## Data availability

The data supporting the findings of this study are available from the corresponding authors upon request. The RNA sequencing data generated in this study have been deposited in the NCBI BioProject database under the accession number PRJNA79183. The mass spectrometry proteomics data were deposited to the ProteomeXchange Consortium via the PRIDE partner repository with the dataset identifier PXD030544. The cryo-EM density maps of the *T. brucei* 80 S ribosome have been deposited in the Electron Microscopy Data Bank (EMDB) under accession numbers EMD-17208 and EMD-17212. Atomic coordinates and structure factors have been deposited in the Protein Data Bank (PDB) under accession codes 8OVA and 8OVE. Source data are provided with this paper.

## Code availability

Bioinformatics scripts used in this study are available at https://github.com/michaelilab/Tb_rRNA_pseudo_translation.

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

## Acknowledgements

This work was supported by a grant from the Israel Science Foundation (1959/20). S.M. holds the David and Inez Myers Chair in RNA silencing of diseases. K.S.R. is supported by the Dean of Faculty Fellowship, Koshland Prize and Sir Charles Clore Postdoctoral Fellowship from the Weizmann Institute. The authors want to thank Noa Aharon-Hefetz and Yitzhak Pilpel for their assistance in preparing the Supplementary Fig. 4.

## Author contributions

K.S.R. and S.M. conceptualization; K.S.R., M.T., A.R., and S.C.C. methodology; K.S.R., A.Bl., M.T. validation; K.S.R., H.M., S.A., A.B.l., B.G., M.T., Y.N., T.D., and E.Z. formal analysis; K.S.R. and A.Ba. investigation; K.S.R. and S.M. writing - original draft; K.S.R., A.Ba., A.Y., and S.M. review and editing; A.Ba., T.I., S.S., R.U., A.Y., and S.M. supervision and project administration; A.Y., and S.M. funding acquisition.

## Competing interests

The authors declare no competing interests.
