## [Peer Review File · Nature Communications]

A single pseudouridine on rRNA regulates ribosome structure and function in the mammalian parasite *Trypanosoma brucei*REVIEWER COMMENTS

Reviewer #1 (Remarks to the Author):

Rajan et al. use multiple methods to describe how pseudouridine modifications on rRNA effect parasite developmental cycles in trypanosomatid parasite development cycles, ultimately arguing that a single pseudouridine modification can alter translation. A strength of the seems to be the combination of HydraPsiSeq and nanopore sequencing, although the discussion of this was somewhat hard to follow as a nonexpert. For example, the description of two types of data and their strengths in the first section of the Results seems somewhat contradictory as someone unfamiliar with the two techniques. This feeling translates to the overall paper: it is hard to follow what the goal of the study was. From the discussion, which touches on pseudouridine in cancer and in *Drosophila* before ending with trypanosomatids (which was nominally the system studied here), it seems that the effect of a single pseudouridine on translation is the ultimate conclusion of the work with the effect in parasites as merely a demonstration. All of this said, I am not an expert in rRNA modifications, pseudouridine, or trypanosomatids so I cannot provide in-depth comments on the importance or context of the work from that perspective. Instead, I was asked to evaluate the mass spectrometry data. I would ask the authors to address the following comments prior to publication:

1. When describing statistical analysis, no specifics are provided that I could find. The authors say that Perseus was used for calculations, which is a great software to use, but no mention of how they performed analyses in Perseus are mentioned. What counts as significant? A t-test with $p < 0.05$? Was multiple testing correction considered (i.e., a permutation-based FDR in Perseus)? Or is there a static fold change and p-value cutoff used instead? These are necessary details to include and also justify why the authors chose this (e.g., why not do multiple testing corrections if they were not done). Without these statements saying things like “the level of 16 proteins was significantly changed” has no meaning.
2. It would be helpful to label hnRNPF/H on the graph in Figure 2G to understand it's fold change with respect to the rest of the proteome.
3. Figure 5B is some what hard to interpret. It is odd that the x-axis goes in reverse order. The two “significant” changes are marked in pink and green, yet the eL38 green dot is far below a $-\log(p\text{-value})$ of ~ 1.3 that would indicate a $p < 0.05$. How is this significant? This further underscores the point above that the authors need to describe what metrics constitute significance.
4. The statement that bioinformatic scripts are available from the corresponding author upon request is unacceptable. These scripts, or a link to a Github account to access they scripts at the very minimum, must be included as part of the manuscript.
5. I commend the authors uploading their data to PRIDE. Can they provide a key to understand what raw file name corresponds to what study. Furthermore, details such as what sample was labeled with light vs heavy dimethyl groups is needed. Currently the methods state “one of the samples” and “the other

sample". All the details necessary to replicate the experiment and analysis need to be provided.

Reviewer #2 (Remarks to the Author):

In the manuscript is entitled "A single pseudouridine on rRNA matters: A specific effect of rRNA modification on translation in the human parasite *Trypanosoma brucei*", authors set to find whether pseudouridylation is developmentally regulated in the life cycle of the parasite *T. brucei*, and whether differential pseudouridine levels at specific positions of the parasite's ribosome can influence translation. This is of interest to the field as it 1) addresses the question whether RNA modifications are developmentally regulated, and 2) addresses the question whether specific RNA modifications contribute the ribosome heterogeneity by altering ribosomal functions. This manuscript complements work previously published by the authors in which they profile Ψ levels and snoRNA expression in the two developmental forms of *T. brucei*, PCF and BSF, and highlighted H69 as hypermodified at 4 out of 5 Ψ positions in the helix (Ψ 518, Ψ 522, Ψ 524 and Ψ 528). However, it adds to their previous work by 1) profiling Ψ levels using additional high-throughput methods, 2) manipulating the Ψ level of specific Ψ sites by deletion and addback of specific snoRNAs, 3) exploring effects of the presence of Ψ on downstream processes (e.g., protein translation, ribosome structure and incorporation of ribosomal proteins).

In the current study, authors profile Ψ levels in a PCF and a BSF strain of the *T. brucei* parasite using HydraPsiSeq and nanopore sequencing, which complement their earlier study (Chikne et al., *Sci. Rep.* 2016). Unlike in the prior publication Ψ profiling was done not only on total RNA but also on polysome-enriched RNA. As indicated by authors, the latter is meant to identify modifications present (or absent) specifically in actively translating ribosomes.

The authors identify a subset of sites differentially modified between total and polysome enriched RNA (most of them showing the same trend in PCF and in BSF), and a subset of sites differentially modified between PCF and BSF, which represent "developmentally regulated" positions. Authors elaborate mostly on positions differentially regulated between total and polysome RNA, although they mention their interest lays in the developmentally regulated sites. Thus, it leaves the reader a bit confused as to what led to choosing the Ψ positions later addressed in the knock-down experiments. Discussed below. Authors choose to focus downstream experiments on two Ψ s occurring in helix 69 (H69): their main target is Ψ 530, which is higher in polysomes vs total RNA in PCF form only. They do not identify Ψ 530 as developmentally regulated. An additional target is Ψ 522, which, according to HydraPsiSeq analysis are hypomodified in polysomes from BSF compared to polysomes from PCF (although in the prior study was considered hypermodified).

They next use a PCF strain to knock-out (KO) each of the two snoRNAs targeting Ψ 530 and Ψ 522, respectively, allowing to test effects of loss of Ψ s in PCF. They find both snoRNAs to be essential, giving rise to knock-down (KD) strains, in which the corresponding position presents reduced Ψ levels. Unlike in their prior publication, where both snoRNAs were co-expressed (with two additional snoRNAs), in this study the two snoRNAs are individually KD or KD and added back, giving more site-specificity to the results.

For Ψ 530, authors show that the KD strain grows slower than the WT and that addback of the snoRNA restores growth to normal and increases levels of Ψ 530. While KD caused no changes in RNA levels of transcripts (checked via RNA-seq) or in global translation, 77 proteins were found to be differentially expressed (checked via MS). Addback of the snoRNA resulted in 16 differentially expressed proteins (the overlap between these two subset of proteins should be discussed). To check whether loss of Ψ 530 affects rRNA structure, authors used DMS-MaPseq on WT and KD strains and quantified the % of mutations at A/C, which correlates to accessibility of the nucleotides in the sequence. They conclude from these studies that KD of the snoRNA targeting Ψ 530 causes global changes in ribosome structure. For Ψ 522, authors show that the KD strain grows slower than the WT and that addback of the snoRNA restores growth to normal.

Authors further claim that KD of the snoRNA directing Ψ 530 (but not Ψ 522) reduced translation of hnRNP F/H and caused a reduction in the presence of the ribosomal protein eS12 in 80S monosomes of PCF strains. Overexpression of eS12 restored protein levels of hnRNP F/H.

Overall, authors focus on the function of Ψ 530, which isn't developmentally regulated in *T.brucei*, but seems hypermodified in polysomes vs total RNA. They suggest that absence of Ψ from position 530 affects the structure of H69 and the ribosome as a whole, thus dislodging eS12 from the ribosome, leading to altered translation of proteins, such as hnRNP F/H.

Authors nicely begin with a wide view on Ψ levels in *T.brucei* ribosomes, but their choice of focusing on Ψ 530 is not entirely clear to this reviewer, as well as the evidence to some of the mechanisms they suggest, including the broad effect on protein translation and ribosome structure, as described below.

Major comments:

1) While authors present Ψ profiling data from 3 whole transcriptome methods and 2 developmental stages of *T.brucei*, there's only a partial overlap between methods for Ψ profiling. Authors choose to focus on Ψ 530 and Ψ 522, which comprise 2 of the 5 Ψ s present in H69. While previous publications, by the authors and others, support the relevance of probing the effect of Ψ in H69 on the function of the ribosome, the motivation behind choosing these specific sites is not strongly supported by the current data. First, there are other Ψ sites outside of H69 that present stronger developmentally regulated differences in modification level. Second, even within H69, Ψ 530 (unlike Ψ 522 and Ψ 524) wasn't highlighted by authors as developmentally regulated. If authors want to focus on highly developmentally regulated Ψ s, they should target more significant hits from the data (e.g., Ψ 1166). If they want to focus on H69 Ψ s (due to additional interest beyond the current screen), they should state their motivation more clearly in the text, and target also more significant Ψ s in H69 (e.g., Ψ 524 or Ψ 518).

2) Authors profile both total and polysome enriched RNA in PCF and BSF form of *T.brucei*. The first part of the manuscript is devoted to a very long description of results from total RNA, measured by 3 whole transcriptomic methods. Authors later mention that a more relevant approach for identifying developmentally regulated positions that may affect translation is to look at Ψ levels of polysome enriched RNA. However, they hardly relate to positions changing in the comparison between BSF and PCF in polysomes in the text (nor in downstream analysis).

3) Statistical significance: there are no p-values attached to any of the Ψ quantifications given in extended tables 1-3. These should be calculated and presented so that the readers can appreciate the

significance (and not only effect size) of the comparisons between conditions. This should be done for both methods of quantification (i.e., HydraPsiSeq and nanopore), and consideration of a site as hyper- or hypo-modified should be made only if the difference between conditions was found significant. This is of importance also since authors use nanopore data, with sometimes very marginal effects (e.g., delta of BSF and PCF for Ψ 522 and Ψ 524, which is -0.01 and -0.02, respectively), to settle discrepancies between psi-Seq and HydraPsiSeq. If these marginal effects are not significant then authors cannot claim the discrepancy is resolved. The same goes to all sites mentioned in line 129: while nanopore “verified the existence” of all 8 sites highlighted in extended table 2, it is not clear to which of these a significant difference between BSF and PCF was seen in the data. In this vein, authors should plot the scores of HydraPsiSeq as a function of scores in nanopore for all detected sites in order to allow a more global inspection of the data.

4) Due to the discrepancy between platforms addressed above, authors should validate the differential level of (at least) Ψ 530 and Ψ 522 using an additional method, such as SCARLET. It would be best if they validate all 5 positions in H69 to settle the discrepancy between the two studies, as they claim modification of this helix is crucial for the function of the ribosome.

5) Polysome profiling patterns were unchanged in WT and snoRNA KD strains, however when looking into proteome of 80S monosomes, authors identified a decrease in eS12 in KD samples. First, why were polysomes, in which authors profiled Ψ sites, not checked? Second, data from yeast (Martin-Villanueva et al., RNA Biology, 2020. doi: 10.1080/15476286.2020.1767951) shows that loss of eS12 results in impaired rRNA processing and changes in observed polysome profiling. Is the lack of a similar phenotype in the snoRNA KD indicative of a different mechanism of action of eS12 in *T.brucei* or is the snoRNA KD not enough to cause a dramatic loss of eS12 and thus no phenotype is observed? To distinguish the two authors should create eS12 deletion strains and check effect on polysome profiling and monosome/polysome proteome composition. Third, authors overexpress eS12, but do not show that upon this overexpression the protein is indeed re-incorporated into the ribosome. They should show this to rule out that the overexpression works via a different route.

6) Authors overexpress eS12 in parental and sKO cells and show results of WB for hnRNP F/H (Fig. 5D) and MS (extended table 7). First, it is not clear why overexpression of eS12 in parental strain causes a decrease in hnRNP F/H protein levels. Authors don't supply any explanation for this. Second, although the WB shows a modulation of protein expression of hnRNP F/H, the protein isn't among the proteins changing in extended table 7. This raises doubts as to the generality of the MS results.

7) Extended data figure 6 shows that eS12 is more abundant in 80S monosomes of BSF vs PCF. Does this correlate with Ψ 530 levels in the two samples (in total and polysome fractions no change in Ψ 530 was detected between forms).

8) What is the overlap between proteins differentially regulated by snoRNA addback and es12 addback. The manuscript implies these would overlap nicely if the hypothesis that snoRNA deficiency affects selective protein synthesis due to eS12 loss from the ribosomal complex.

9) Extended table 4 and 5 – add gene symbol name to the table, such that readers could identify proteins from figures in the table. Of note, this reviewer could not find some of the proteins mentioned in figure 2H in the supplementary table, including hnRNP H/F.

10) Figure 4: authors should discriminate between changes in mismatch % that come from technical reasons (e.g., differences in DMS efficiency in the in vivo incubation stage), which would cause a global

shift in mutational signal in all sites, and between changes in mismatch % that result from changes in structure (and would thus most likely be present in a subset of sites). It would also help if data from Figure 4D-E was presented as in Figure 4C, i.e., comparing the mutation % of each position in WT vs KO. Another possibility is to show DMS-MaPSeq data they generated for non-ribosomal RNAs, and see if there the signal is stable between conditions (If other types of RNA show similar changes in % mismatch this would support a non-specific effect of the genetic manipulation). In the current display of the DMS results, this reviewer is not convinced that DMS results suggest a marked change in structure. Perhaps it would help if these results were supported by additional methods e.g., SHAPE-seq or preferably cryo-EM.

11) The authors shift from profiling total RNA to polysome-enriched RNA to monosomes, in each case checking different outputs (modification level, general RNA and protein abundance, specific incorporation of proteins to the ribosome, ribosome structure etc). It is important that authors be consistent with their profiling to indicate that effects they see in one experiment indeed reflect what was found in the others (e.g., is showing Ψ level differ in polysome, why profile proteins bound to ribosomes in monosome?). If the shift is relevant for their hypothesis authors should emphasize in the text why a different fraction was relevant for the specific output that was measured and how it fits to prior experiments.

Minor comments:

- 1) Authors compare 3 methods for Ψ mapping, which show partial overlap between them. While authors partially address this, it would be helpful if they present a panel, e.g., a pie chart, of the overlap between platforms. It would help if such a plot will show only statistically significant changing positions of each comparison (e.g., polysome BSF vs PCF, total vs polysome in PCF etc).
- 2) Abstract line 30: "...snoRNA guiding Ψ 530 in helix 69 (H69) altered the...". Change as in underlined.
- 3) Abstract line 33: "...overexpressing the ribosomal protein eS12". Change as in underlined.
- 4) Line 67: Reference 20 and 21 are the same
- 5) In the introduction, explain more about H69 and what is known about it in the context of Ψ , snoRNA and ribosomal functions.
- 6) Figure 1A – highlight the statistically significant sites (authors refer to 5 in the text. Are there more?). Be clear about which are relevant for the addressed hypothesis (i.e., developmentally regulated? Present mostly in active ribosomes?).
- 7) Lines 133-136: please indicate the Ψ position when mentioning the corresponding location in the ribosome (e.g., "...one site is located in H69 domain (Ψ 518), one...").
- 8) Lines 173-180: authors describe the RNA-seq and only then the proteome analysis. However, in describing the RNA-seq results they refer to results of the proteomic analysis ("especially for...de-regulated proteins"). Either delete the end of the sentence in line 175 or describe the proteomic data before the RNA-seq data.
- 9) Line 181: please add the underlined text: "antibodies specific to 7 of the identified proteins."
- 10) Figure 2H: explain the rationale of choosing the 7 genes for WB validation (most highly changing? Most significant?). Are all these 7 proteins appearing in the relevant supplemental tables? It would help if authors highlight in figure 2G the proteins that were subjected for further investigation via WB. In the WB, indicate which gene was meant to serve as a loading control. If all 7 are meant to be changing, add

the WB of a control, non-changing protein to the blot. Also, out of these 7 genes, only one (hnRNP F/H) showed reduced expression upon snoRNA KO. This low correlation between results of Mass-spec and results of WB should be addressed.

11) Figure 2Bi and extended data figure 3Bi are the same.

12) Add p-values to Figure 2C, Figure 3Biii, Figure 4D/E, extended data figure 5iii, Extended Tables.

13) Previous work by the authors showed that hnRNP F/H is developmentally regulated in this system (Gupta et al., NAR 2013). Did the gene come up in the proteomic analysis described in extended figure 5 as well? If not, authors should mention this discrepancy and its potential source(s).

14) Line 186: authors state that 60 of the 77 differentially expressed proteins upon snoRNA KO are developmentally regulated. First, is this significantly enriched compared to the fraction of developmentally regulated proteins in snoRNA independent proteins? Second, if Ψ 530 is not differentially regulated (as it wasn't found significantly changing in any BSF/PCF comparison), why should the proteins changing upon its decrease be developmentally regulated?

15) Extended data figure 5: ii) please re-check p-values. Based on the plot and its error bars it is peculiar that WT OE had a more significant result than Mut OE. iii) please add p-values. It isn't clear what is the difference between this panel and Figure 3Biii. Are they both part of the same experiment?

16) Line 215: "The H69 protein carries 5 Ψ s. To assess if each of the...". Change "protein" to "helix", change "each" to "additional", as not all Ψ s were examined.

17) Authors conducted quantification of proteins upon KO or KO+addback of the snoRNA. Each condition resulted in a few differential proteins. The authors should indicate the overlap between these two measurements. Ideally, one would expect the KO+addback to restore effect on proteins found in the KO samples.

18) Extended data figure 3Di is missing a loading control.

19) Figure 4C: it is hard to distinguish between base type (circle or triangle). Better to color-code the base types and not the mutation mismatch. To indicate mutation % authors can draw dashed lines at the relevant cutoff values.

20) Line 249: please add reference to the relevant extended table showing data from the MS experiment presented in Figure 5B.

21) Extended data figure 1D : Numbering of the positions in extended data figure 1D isn't consistent with the rest of the paper. Please unify. Furthermore, if not all positions shown in the plot are considered statistically significant, please indicate which sites are significant by coloring the relevant dots in a different color.

22) As snoRNAs were found essential, the generated strains are actually KD and not KO strains. Please amend text and figures accordingly.

23) The authors further relate to data they previously published using Ψ -seq (Chikne et al., Sci. Rep. 2016). They mention, "Notably, all these hyper-modified sites in BSF were previously detected by Ψ -seq". While authors relate to 5 hyper-modified sites identified by HydraPsiSeq, Ψ -seq identified only 4 of these, as in the earlier publication Ψ 1167 and not Ψ 1166 was identified as hyper-modified. Authors should clarify this point.

24) Authors show that addback of WT but not mutant snoRNA TB11Cs6H1 (targeting Ψ 530) increases protein levels of hnRNP F/H and thus relate it to a regulation of the protein's translation. How does these addbacks effect RNA level of hnRNP F/H? Does the effect on protein levels stem from increased

translation or reduced degradation of the protein?

25) When addressing the results of eS12 in the discussion, refer also to prior work in yeast (Martin-Villanueva et al., RNA Biology, 2020. doi: 10.1080/15476286.2020.1767951) and human (Brumwell et al., RNA, 2020, doi: 10.1261/rna.070318.119.).

26) Are the snoRNAs of Ψ 530 and Ψ 522 differentially expressed between BSF and PCF in a way that could explain the difference in Ψ levels? Would overexpression of TB11Cs6H1 in BSF increase levels of Ψ 530 to those of PCF and would that result in change in eS12 presence in the ribosomes?

27) Please indicate on which fraction of RNA was DMS-MaPSeq conducted (polysome/monosome etc).

Reviewer #3 (Remarks to the Author):

The manuscript by Rajan et al describes application of three RNA-Seq based methods to identify sites of pseudouridine modification in ribosomal RNAs of parasitic protozoan *Trypanosoma brucei*. The quality of these datasets is high and the most prominent hypo or hyper modified positions are largely consistent among approaches that detect chemical modification, strand breakdown or pausing at pseudouridine site. The focus then shifts to a specific position 530 and the TB11Cs6H1 snoRNA, previously characterized by the same group, that presumably guides modification at this position. It is well established that PTC is heavily modified in probably all organisms, which brings at least two-fold burden on establishing a critical role of any given position. Developmental changes in modification prevalence are indicative of biological relevance but a formal connection from particular snoRNA to modification site to (presumed) ribosome structural change to preferential translation of some mRNAs is an endeavor that requires efficient genetic tools to start with. Here, snoRNA connection to modification position rests on a single allele KO and the effects are borderline. If conditional snoRNA expression is possible, there is no apparent reason for not establishing a conditional null KO, be it by CRISPR or recombination. This would allow monitoring effects in a time-resolved manner and hopefully generate more apparent impact on selective mRNA translation or ribosome composition, or any other aspect of trypanosome metabolism. Limiting the observed phenotype to a single LSU modification is problematic considering the report of snRNA modifications by TB11Cs6H1 snoRNA. The conjecture between a single modification loss in LSU and S12 reduction is unclear just like the reasons for snoRNA sKO compensation by S12 overexpression. These are interesting preliminary findings on their own but give limited mechanistic insights at this point. Certainly worth pursuing further.

Enclosed please find a revised version of our study entitled “A single pseudouridine on rRNA matters: A specific effect of rRNA modification on translation in the mammalian parasite *Trypanosoma brucei*”. We spent several months to revise the manuscript because we decided to strengthen our claims by examining the atomic structure of the ribosomes of the TB11Cs6H1 sKO in order to verify that indeed the lack of a single Ψ modification resulted in ribosomes lacking eS12. We therefore teamed with the group of Prof. Ada Yonath to determine the cryo-EM structure of ribosomes from the parental strain and the sKO. In this study, we present a high-resolution refined structure, compared with the lower resolution structure published by Hashem *et al.*, Nature (2013), and show all the rRNA modifications. Our responses to the reviewers’ comments are summarized below:

Reviewer #1 (Remarks to the Author):

Rajan et al. use multiple methods to describe how pseudouridine modifications on rRNA effect parasite developmental cycles in trypanosomatid parasite development cycles, ultimately arguing that a single pseudouridine modification can alter translation. A strength of the seems to be the combination of HydraPsiSeq and nanopore sequencing, although the discussion of this was somewhat hard to follow as a nonexpert. For example, the description of two types of data and their strengths in the first section of the Results seems somewhat contradictory as someone unfamiliar with the two techniques. This feeling translates to the overall paper: it is hard to follow what the goal of the study was. From the discussion, which touches on pseudouridine in cancer and in Drosophila before ending with trypanosomatids (which was nominally the system studied here), it seems that the effect of a single pseudouridine on translation is the conclusion of the work with the effect in parasites as merely a demonstration. All of this said, I am not an expert in rRNA modifications, pseudouridine, or trypanosomatids so I cannot provide in-depth comments on the importance or context of the work from that perspective. Instead, I was asked to evaluate the mass spectrometry data. I would ask the authors to address the following comments prior to publication.

The purpose of the study was to understand if a single modification could impact translation, and if so, to determine whether the effect on translation is general or specific. The differential modification observed between the two parasite stages may suggest that this modification could contribute to the proteome changes that take place while cycling between the two hosts. To examine this hypothesis, it was necessary to demonstrate changes occurring due to the depletion of the snoRNAs that guide such developmentally regulated Ψ positions. We focused on Ψ s located in H69 because these are universally conserved and are guided by single copy genes encoding a single snoRNA, and thus amenable to CRISPR-Cas9 manipulation. H69 Ψ s are developmentally regulated, and their levels vary between total and polysomal RNA. In the previous version of the manuscript, we studied the function of two such snoRNAs and only one of them (TB11Cs6H1) demonstrated a clear phenotype impacting translation. In the current version we included functional analyses on the additional snoRNAs that guide on H69, as requested. We were not able to obtain even a sKO for the remaining three snoRNAs. We therefore used the construct from our previous study (Chikne *et al.*, 2016) to overexpress these snoRNAs, but in each construct one snoRNA was mutated. Analysing the proteome of such mutants allowed us to identify specific translation defects, suggesting that these snoRNAs also contribute to regulation of translation.

1. When describing statistical analysis, no specifics are provided that I could find. The authors say that Perseus was used for calculations, which is a great software to use, but no mention of how they performed analyses in Perseus are mentioned. What counts as significant? A t-test

with $p < 0.05$? Was multiple testing correction considered (i.e., a permutation-based FDR in Perseus)? Or is there a static fold change and p -value cutoff used instead? These are necessary details to include and also justify why the authors chose this (e.g., why not do multiple testing corrections if they were not done). Without these statements saying things like “the level of 16 proteins was significantly changed” has no meaning.

As requested, we now provide a more detailed description of the statistical analysis in the Methods section. We chose proteins that were increased or decreased by 1.4-fold and have a p -value of less than 0.05. For TB11Cs6H1 overexpression, we provide data from two clonal replicates, and chose those proteins that are altered (increased or decreased by at least 1.4-fold and p values less than 0.05).

2. It would be helpful to label hnRNPF/H on the graph in Figure 2G to understand its fold change with respect to the rest of the proteome.

We marked in Fig. 2G all proteins shown in the western blot [Fig. 2H] (except hnRNP F/H, RBP10 and MTAP). hnRNP F/H and RBP10 were shown to be developmentally regulated and highly expressed in bloodstream form, and hence were not detected in the proteome because these proteins are not abundant enough in procyclic stage parasites to be detected by MS.

3. Figure 5B is somewhat hard to interpret. It is odd that the x -axis goes in reverse order. The two “significant” changes are marked in pink and green, yet the eL38 green dot is far below a $-\log(p\text{-value})$ of ~ 1.3 that would indicate a $p < 0.05$. How is this significant? This further underscores the point above that the authors need to describe what metrics constitute significance.

We replotted the graph with the x -axis in the correct orientation. eL38 was increased in two replicates, while eS12 was increased in all three replicates. In the revised version, we highlight only eS12.

4. The statement that bioinformatic scripts are available from the corresponding author upon request is unacceptable. These scripts, or a link to a Github account to access they scripts at the very minimum, must be included as part of the manuscript.

The bioinformatics scripts were deposited in Github, and the link is now provided in manuscript.

5. I commend the authors uploading their data to PRIDE. Can they provide a key to understand what raw file name corresponds to what study. Furthermore, details such as what sample was labeled with light vs heavy dimethyl groups is needed. Currently the methods state “one of the samples” and “the other sample”. All the details necessary to replicate the experiment and analysis need to be provided.

Supplementary Table 15 was added and includes names of the samples, description of samples, and heavy and light label.

Reviewer #2 (Remarks to the Author):

In the manuscript is entitled "A single pseudouridine on rRNA matters: A specific effect of rRNA modification on translation in the human parasite *Trypanosoma brucei*", authors set to

find whether pseudouridylation is developmentally regulated in the life cycle of the parasite T. brucei, and whether differential pseudouridine levels at specific positions of the parasite's ribosome can influence translation. This is of interest to the field as it 1) addresses the question whether RNA modifications are developmentally regulated, and 2) addresses the question whether specific RNA modifications contribute the ribosome heterogeneity by altering ribosomal functions. This manuscript complements work previously published by the authors in which they profile Ψ levels and snoRNA expression in the two developmental forms of T. brucei, PCF and BSF, and highlighted H69 as hypermodified at 4 out of 5 Ψ positions in the helix (Ψ 518, Ψ 522, Ψ 524 and Ψ 528). However, it adds to their previous work by 1) profiling Ψ levels using additional high-throughput methods, 2) manipulating the Ψ level of specific Ψ sites by deletion and addback of specific snoRNAs, 3) exploring effects of the presence of Ψ on downstream processes (e.g., protein translation, ribosome structure and incorporation of ribosomal proteins).

In the current study, authors profile Ψ levels in a PCF and a BSF strain of the T. brucei parasite using HydraPsiSeq and nanopore sequencing, which complement their earlier study (Chikne et al., Sci. Rep. 2016). Unlike in the prior publication Ψ profiling was done not only on total RNA but also on polysome-enriched RNA. As indicated by authors, the latter is meant to identify modifications present (or absent) specifically in actively translating ribosomes. The authors identify a subset of sites differentially modified between total and polysome enriched RNA (most of them showing the same trend in PCF and in BSF), and a subset of sites differentially modified between PCF and BSF, which represent “developmentally regulated” positions. Authors elaborate mostly on positions differentially regulated between total and polysome RNA, although they mention their interest lays in the developmentally regulated sites. Thus, it leaves the reader a bit confused as to what led to choosing the Ψ positions later addressed in the knock-down experiments. Discussed below. Authors choose to focus downstream experiments on two Ψ s occurring in helix 69 (H69): their main target is Ψ 530, which is higher in polysomes vs total RNA in PCF form only. They do not identify Ψ 530 as developmentally regulated. An additional target is Ψ 522, which, according to HydraPsiSeq analysis are hypomodified in polysomes from BSF compared to polysomes from PCF (although in the prior study was considered hypermodified). They next use a PCF strain to knock-out (KO) each of the two snoRNAs targeting Ψ 530 and Ψ 522, respectively, allowing to test effects of loss of Ψ s in PCF. They find both snoRNAs to be essential, giving rise to knock-down (KD) strains, in which the corresponding position presents reduced Ψ levels. Unlike in their prior publication, where both snoRNAs were co-expressed (with two additional snoRNAs), in this study the two snoRNAs are individually KD or KD and added back, giving more site-specificity to the results. For Ψ 530, authors show that the KD strain grows slower than the WT and that addback of the snoRNA restores growth to normal and increases levels of Ψ 530. While KD caused no changes in RNA levels of transcripts (checked via RNA-seq) or in global translation, 77 proteins were found to be differentially expressed (checked via MS). Addback of the snoRNA resulted in 16 differentially expressed proteins (the overlap between these two subset of proteins should be discussed). To check whether loss of Ψ 530 affects rRNA structure, authors used DMS-MaPseq on WT and KD strains and quantified the % of mutations at A/C, which correlates to accessibility of the nucleotides in the sequence. They conclude from these studies that KD of the snoRNA targeting Ψ 530 causes global changes in ribosome structure. For Ψ 522, authors show that the KD strain grows slower than the WT and that addback of the snoRNA restores growth to normal. Authors further claim that KD of the snoRNA directing Ψ 530 (but not Ψ 522) reduced translation of hnRNP F/H and caused a reduction in the presence of the ribosomal protein

eS12 in 80S monosomes of PCF strains. Overexpression of eS12 restored protein levels of hnRNP F/H.

*Overall, authors focus on the function of Ψ 530, which isn't developmentally regulated in *T.brucei*, but seems hypermodified in polysomes vs total RNA. They suggest that absence of Ψ from position 530 affects the structure of H69 and the ribosome as a whole, thus dislodging eS12 from the ribosome, leading to altered translation of proteins, such as hnRNP F/H. Authors nicely begin with a wide view on Ψ levels in *T.brucei* ribosomes, but their choice of focusing on Ψ 530 is not entirely clear to this reviewer, as well as the evidence to some of the mechanisms they suggest, including the broad effect on protein translation and ribosome structure, as described below.*

As indicated above we chose H69 because of its functional importance and location at the inter-subunit bridge. We attempted to delete all of the snoRNAs guiding Ψ on H69 by CRISPR-Cas9, but failed to attain even a sKO for three of them. We were able to obtain sKO for two snoRNAs, as described in the manuscript. However, we now provide functional data on the remaining snoRNAs guiding Ψ on H69 using combinatorial overexpression. Note that in yeast, the role of Ψ on H69 was studied (Liang *et al.*, Mol. Cell 2006) and the phenotype obtained was different. Only combinations deleting three snoRNAs resulted in a phenotype. However, in trypanosomes this domain seems to be essential for the function of the ribosome and it was not even possible to get a sKO for all snoRNA guiding on this domain.

Major comments:

*1) While authors present Ψ profiling data from 3 whole transcriptome methods and 2 developmental stages of *T.brucei*, there's only a partial overlap between methods for Ψ profiling. Authors choose to focus on Ψ 530 and Ψ 522, which comprise 2 of the 5 Ψ s present in H69. While previous publications, by the authors and others, support the relevance of probing the effect of Ψ in H69 on the function of the ribosome, the motivation behind choosing these specific sites is not strongly supported by the current data. First, there are other Ψ sites outside of H69 that present stronger developmentally regulated differences in modification level. Second, even within H69, Ψ 530 (unlike Ψ 522 and Ψ 524) wasn't highlighted by authors as developmentally regulated. If authors want to focus on highly developmentally regulated Ψ s, they should target more significant hits from the data (e.g., Ψ 1166). If they want to focus on H69 Ψ s (due to additional interest beyond the current screen), they should state their motivation more clearly in the text, and target also more significant Ψ s in H69 (e.g., Ψ 524 or Ψ 518).*

Our study has two purposes, one is to provide a comprehensive analysis of the level of Ψ during cycling of the parasite. This is now addressed by adding a fourth method of tandem LC-MS. The second purpose is to explore the function of Ψ on important functional domains such as H69, in which the role of Ψ s was studied in yeast and bacteria. The reason for not studying the function of other snoRNAs guiding developmentally regulated sites is as follows. As stated above, CRISPR-Cas9 KO could only be used on single copy genes. However, most of the H/ACA snoRNAs guiding developmentally regulated Ψ s are encoded by clusters containing both C/D and H/ACA RNAs, and accordingly, it is not possible to manipulate their expression by this method. Note that snoRNAi is not as efficient for silencing H/ACA as it is for C/D snoRNAs (Liang *et al.*, PNAS 2003). We now provide functional data regarding the remaining snoRNAs guiding Ψ on H69 using combinatorial overexpression.

*2) Authors profile both total and polysome enriched RNA in PCF and BSF form of *T.brucei*. The first part of the manuscript is devoted to a very long description of results from total RNA, measured by 3 whole transcriptomic methods. Authors later mention that a more relevant*

approach for identifying developmentally regulated positions that may affect translation is to look at Ψ levels of polysome enriched RNA. However, they hardly relate to positions changing in the comparison between BSF and PCF in polysomes in the text (nor in downstream analysis).

It is true that we don't use all of our data in this study (we plan to use it in the future) but it is very important to report, especially regarding the role of Ψ 530. Note that no such analysis on polysomal rRNA Ψ was performed previously in any other system. Our results clearly indicate that there is heterogeneity even among the translating ribosomes.

3) Statistical significance: there are no p-values attached to any of the Ψ quantifications given in extended tables 1-3. These should be calculated and presented so that the readers can appreciate the significance (and not only effect size) of the comparisons between conditions. This should be done for both methods of quantification (i.e., HydraPsiSeq and nanopore), and consideration of a site as hyper- or hypo-modified should be made only if the difference between conditions was found significant. This is of importance also since authors use nanopore data, with sometimes very marginal effects (e.g., delta of BSF and PCF for Ψ 522 and Ψ 524, which is -0.01 and -0.02, respectively), to settle discrepancies between psi-Seq and HydraPsiSeq. If these marginal effects are not significant then authors cannot claim the discrepancy is resolved. The same goes to all sites mentioned in line 129: while nanopore "verified the existence" of all 8 sites highlighted in extended table 2, it is not clear to which of these a significant difference between BSF and PCF was seen in the data. In this vein, authors should plot the scores of HydraPsiSeq as a function of scores in nanopore for all detected sites in order to allow a more global inspection of the data.

P-values were calculated for all HydraPsiSeq data, and only $p < 0.05$ was considered significant. The p-values are indicated in the main figure and supplementary file. We also performed tandem LC-MS to quantify the Ψ sites. A comparison of all four methods is now provided in Fig 1B. We provide a graph comparing pairs of each of the methods.

4) Due to the discrepancy between platforms addressed above, authors should validate the differential level of (at least) Ψ 530 and Ψ 522 using an additional method, such as SCARLET. It would be best if they validate all 5 positions in H69 to settle the discrepancy between the two studies, as they claim modification of this helix is crucial for the function of the ribosome.

We performed tandem LC-MS to quantify all the Ψ sites. All four methods support the hypermodification of three sites, SSU_2116, SSU_2146 and LSU α _1481. The Ψ on H69 LSU β _518 was verified as hypomodified by three different methods, the remaining sites were unchanged except in the Ψ -seq.

5) Polysome profiling patterns were unchanged in WT and snoRNA KD strains, however when looking into proteome of 80S monosomes, authors identified a decrease in eS12 in KD samples. First, why were polysomes, in which authors profiled Ψ sites, not checked? Second, data from yeast (Martin-Villanueva et al., RNA Biology, 2020. doi: 10.1080/15476286.2020.1767951) shows that loss of eS12 results in impaired rRNA processing and changes in observed polysome profiling. Is the lack of a similar phenotype in the snoRNA KD indicative of a different mechanism of action of eS12 in T.brucei or is the snoRNA KD not enough to cause a dramatic loss of eS12 and thus no phenotype is observed? To distinguish the two authors should create eS12 deletion strains and check effect on polysome profiling and monosome/polysome proteome composition. Third, authors overexpress eS12, but do not show that upon this

overexpression the protein is indeed re-incorporated into the ribosome. They should show this to rule out that the overexpression works via a different route.

The reason why we did not determine the composition of polysomal ribosomal proteins is because of the difficulties in obtaining sufficient material. As for the function of eS12, we silenced eS12 using a stem-loop RNAi construct and noticed no significant difference in 80S or polysome formation, despite efficient silencing. These data are presented in Figure 5E-F. To determine the incorporation of eS12 to the ribosomes, we examined by sucrose gradient fractionation whether the MYC tagged eS12 is incorporated into the ribosome (Fig 5Di). eS12 was found in 40S but mostly in 80S monosomes. We next verified the levels of eS12 by mass-spec of pure ribosomes. The result, shown in extended Fig. 8, indicates that upon eS12 addback, its level is increased by 2.67-fold.

6) Authors overexpress eS12 in parental and sKO cells and show results of WB for hnRNP F/H (Fig. 5D) and MS (extended table 7). First, it is not clear why overexpression of eS12 in parental strain causes a decrease in hnRNP F/H protein levels. Authors don't supply any explanation for this. Second, although the WB shows a modulation of protein expression of hnRNP F/H, the protein isn't among the proteins changing in extended table 7. This raises doubts as to the generality of the MS results.

hnRNP F/H and RBP10, as stated above, are not abundant enough to be detected in mass-spec as these are developmentally regulated and more abundant in the BSF. We do not know how eS12 overexpression affects hnRNP F/H in the parental cell. The translation of this protein is very sensitive to any change in ribosomes; hence we cannot rule out the possibility that the tagged protein is not 100% functional.

7) Extended data figure 6 shows that eS12 is more abundant in 80S monosomes of BSF vs PCF. Does this correlate with Ψ 530 levels in the two samples (in total and polysome fractions no change in Ψ 530 was detected between forms).

Indeed, eS12 is more abundant in BSF based on ribosome profiling. However, Ψ 530 is not identified as developmentally regulated by any mapping method.

8) What is the overlap between proteins differentially regulated by snoRNA addback and es12 addback. The manuscript implies these would overlap nicely if the hypothesis that snoRNA deficiency affects selective protein synthesis due to eS12 loss from the ribosomal complex.

We provided a proof of concept using hnRNPF/H as a reporter for compensating the phenotype of loss of the snoRNA by eS12 addback. However, the changes in the rRNA structure may cause other defects in addition to dislodging eS12. Note that we also observed the loss of eL38 in two out of the three replicates. We cannot rule out the possibility that some defects emerge from the TB11Cs6H1 guiding on mRNAs or other ncRNAs (this is now addressed in the Discussion).

9) Extended table 4 and 5 – add gene symbol name to the table, such that readers could identify proteins from figures in the table. Of note, this reviewer could not find some of the proteins mentioned in figure 2H in the supplementary table, including hnRNP H/F.

We added the gene symbols. Note that hnRNP F/H and RBP10 are not sufficiently abundant to be detected in mass-spec as these are developmentally regulated and more abundant in the BSF.

10) Figure 4: authors should discriminate between changes in mismatch % that come from technical reasons (e.g., differences in DMS efficiency in the *in vivo* incubation stage), which would cause a global shift in mutational signal in all sites, and between changes in mismatch % that result from changes in structure (and would thus most likely be present in a subset of sites). It would also help if data from Figure 4D-E was presented as in Figure 4C, i.e., comparing the mutation % of each position in WT vs KO. Another possibility is to show DMS-MaPSeq data they generated for non-ribosomal RNAs, and see if there the signal is stable between conditions (If other types of RNA show similar changes in % mismatch this would support a non-specific effect of the genetic manipulation). In the current display of the DMS results, this reviewer is not convinced that DMS results suggest a marked change in structure. Perhaps it would help if these results were supported by additional methods e.g., SHAPE-seq or preferably cryo-EM.

We decided to remove the DMS-MaPSeq analysis from the manuscript and to provide a more direct analysis of structural changes of the ribosomes present in TB11Cs6H1 sKO cells compared to the parental strain, using cryo-EM. Indeed, structural changes in rRNA were observed in different domains outside H69, supporting our hypothesis that the lack of Ψ 530 induced structural changes in the ribosome.

11) The authors shift from profiling total RNA to polysome-enriched RNA to monosomes, in each case checking different outputs (modification level, general RNA and protein abundance, specific incorporation of proteins to the ribosome, ribosome structure etc). It is important that authors be consistent with their profiling to indicate that effects they see in one experiment indeed reflect what was found in the others (e.g., is showing Ψ level differ in polysome, why profile proteins bound to ribosomes in monosome?). If the shift is relevant for their hypothesis authors should emphasize in the text why a different fraction was relevant for the specific output that was measured and how it fits to prior experiments.

Our goal was to determine if there is a difference in Ψ between total RNA and polysomal RNA, since polysomal RNA is extracted from active ribosomes that seem to have a distinct repertoire of Ψ s. Indeed, this was the case. Although as suggested by the reviewer, it would have been optimal to determine the RP composition in polysomal RNA, but we could not obtain enough material for this analysis. We then decided to determine the composition of monosomes. Since all monosomes lacked eS12, we believe it is likely that polysomal ribosomes share the same defect. We now explain our rationale in the revised manuscript.

Minor comments:

1) Authors compare 3 methods for Ψ mapping, which show partial overlap between them. While authors partially address this, it would be helpful if they present a panel, e.g., a pie chart, of the overlap between platforms. It would help if such a plot will show only statistically significant changing positions of each comparison (e.g., polysome BSF vs PCF, total vs polysome in PCF etc).

We added a comparative diagram with data from only statistically significant sites based on the *p*-value that we obtained from HydraPseq (Fig. 1B, Supplementary Table 2).

2) Abstract line 30: “...snoRNA guiding Ψ 530 in helix 69 (H69) altered the...”. Change as in underlined.

We corrected according to the referee's suggestion.

3) Abstract line 33: “...overexpressing the ribosomal protein eS12”. Change as in underlined.

We corrected the mistake.

4) Line 67: Reference 20 and 21 are the same

We corrected the mistake.

5) In the introduction, explain more about H69 and what is known about it in the context of psi, snoRNA and ribosomal functions.

We added information regarding Ψ on H69 from yeast and bacteria, as well as describing the function of H69 in translation based on recently published papers.

6) Figure 1A – highlight the statistically significant sites (authors refer to 5 in the text. Are there more?). Be clear about which are relevant for the addressed hypothesis (i.e., developmentally regulated? Present mostly in active ribosomes?).

In the current version, we present only the statistically significant sites based on p-value obtained from HydraPsiSeq (4 shared among 4 methods, and 5 among three methods).

7) Lines 133-136: please indicate the Ψ position when mentioning the corresponding location in the ribosome (e.g., “...one site is located in H69 domain (Ψ 518), one...”).

We now mention all the site designations in the text.

8) Lines 173-180: authors describe the RNA-seq and only then the proteome analysis. However, in describing the RNA-seq results they refer to results of the proteomic analysis (“especially for...de-regulated proteins”). Either delete the end of the sentence in line 175 or describe the proteomic data before the RNA-seq data.

We re-wrote the section and describe the RNA-seq before the proteome analysis.

9) Line 181: please add the underlined text: “antibodies specific to 7 of the identified proteins.”

We now mention the antibody specificity in the main text.

10) Figure 2H: explain the rationale of choosing the 7 genes for WB validation (most highly changing? Most significant?). Are all these 7 proteins appearing in the relevant supplemental tables? It would help if authors highlight in figure 2G the proteins that were subjected for further investigation via WB. In the WB, indicate which gene was meant to serve as a loading control. If all 7 are meant to be changing, add the WB of a control, non-changing protein to the blot. Also, out of these 7 genes, only one (hnRNP F/H) showed reduced expression upon

snoRNA KO. This low correlation between results of Mass-spec and results of WB should be addressed.

The seven proteins were chosen because these were previously studied by our group and the antibodies were generated by us. Three proteins among the seven were not sufficiently abundant to be detected by MS. Expression of the proteins present in MS was not changed, as seen by the WB. We indicated the four proteins studied by WB in Fig. 2G. These four proteins can be considered a loading control.

11) *Figure 2Bi and extended data figure 3Bi are the same.*

Correct. In Extended Figure 4Bi, we indicated the position of PCR primers used to verify the integration of the resistance genes.

12) *Add p-values to Figure 2C, Figure 3Biii, Figure 4D/E, extended data figure 5iii, Extended Tables.*

We added *p*-values to all these Figures.

13) *Previous work by the authors showed that hnRNP F/H is developmentally regulated in this system (Gupta et al., NAR 2013). Did the gene come up in the proteomic analysis described in extended figure 5 as well? If not, authors should mention this discrepancy and its potential source(s).*

As described in our previous publication (Gupta et al., *Nucleic Acid. Res.* 2013), hnRNPF/H is poorly expressed in PCF. We now indicate this fact in the text.

14) *Line 186: authors state that 60 of the 77 differentially expressed proteins upon snoRNA KO are developmentally regulated. First, is this significantly enriched compared to the fraction of developmentally regulated proteins in snoRNA independent proteins?*

Yes, only 10.6% of the *T. brucei* proteome is developmentally regulated, and therefore, the 85.7% of the affected protein in our dataset is significant. We indicate this in the text.

Second, If Ψ530 is not differentially regulated (as it wasn't found significantly changing in any BSF/PCF comparison), why should the proteins changing upon its decrease be developmentally regulated?

As can be seen, Ψ530 is significantly increased in the polysomes. It is possible that translation of developmentally regulated genes is more sensitive to any change in the ribosome. We currently do not know what properties are shared by all these mRNAs resulting in their altered translation.

15) *Extended data figure 5: ii) please re-check p-values. Based on the plot and its error bars it is peculiar that WT OE had a more significant result than Mut OE. iii) please add p-values. It isn't clear what is the difference between this panel and Figure 3Biii. Are they both part of the same experiment?*

The previous *p*-value was in error, and it was corrected.

16) Line 215: “The H69 protein carries 5 Ψs. To assess if each of the...”. Change “protein” to “helix, change “each” to “additional”, as not all Ψs were examined.

Correct. In the current version we added data regarding the four out of five Ψs on H69.

17) Authors conducted quantification of proteins upon KO or KO+addback of the snoRNA. Each condition resulted in a few differential proteins. The authors should indicate the overlap between these two measurements. Ideally, one would expect the KO+addback to restore effect on proteins found in the KO samples.

In fact, we showed only WB supporting the effect of losing hnRNP F/H.

18) Extended data figure 3Di is missing a loading control.

ZC3H41, PTB1, and HSP83 serve as loading controls. This is now indicated in the Figure legend.

19) Figure 4C: it is hard to distinguish between base type (circle or triangle). Better to color-code the base types and not the mutation mismatch. To indicate mutation % authors can draw dashed lines at the relevant cutoff values.

We removed these data in the revised manuscript, and instead, added cryo-EM data.

20) Line 249: please add reference to the relevant extended table showing data from the MS experiment presented in Figure 5B.

The relevant data were added to Extended Table 9.

21) Extended data figure 1D : Numbering of the positions in extended data figure 1D isn't consistent with the rest of the paper. Please unify. Furthermore, if not all positions shown in the plot are considered statistically significant, please indicate which sites are significant by coloring the relevant dots in a different color.

We indicated its location on the ribosome subunit. This result was used as supportive evidence for the HydraPsiSeq and was performed only once. We now note this in the text.

22) As snoRNAs were found essential, the generated strains are actually KD and not KO strains. Please amend text and figures accordingly.

In the field of trypanosome research, KD is used to denote RNAi silencing. We therefore prefer to refer to our cell lines as sKO.

23) The authors further relate to data they previously published using Ψ-seq (Chikne et al., *Sci. Rep.* 2016). They mention, “Notably, all these hyper-modified sites in BSF were previously detected by Ψ-seq”. While authors relate to 5 hyper-modified sites identified by HydraPsiSeq, Ψ-seq identified only 4 of these, as in the earlier publication Ψ1167 and not Ψ1166 was identified as hyper-modified. Authors should clarify this point.

Previously, in Chikne et al., Ψ 1167 was identified by Ψ -seq and assigned as hypermodified, whereas HydraPseq detected only Ψ 1166. However, tandem LC-MS identified both positions as hypermodified.

24) Authors show that addback of WT but not mutant snoRNA *TB11Cs6H1* (targeting Ψ 530) increases protein levels of hnRNP F/H and thus relate it to a regulation of the protein's translation. How does these addbacks effect RNA level of hnRNP F/H?

We did not evaluate the level of hnRNPF/H because we did not see a reason why this will happen and why it should be altered. We did not see any change in mRNA level of hnRNP F/H in the sKO cells by RNA-seq and Northern analysis.

Does the effect on protein levels stem from increased translation or reduced degradation of the protein?

Based on the fact that we observed no change in the level of mRNA in the sKO we suggest that the effect is on translation, and also because our alteration was on the rRNA.

25) When addressing the results of *eS12* in the discussion, refer also to prior work in yeast (Martin-Villanueva et al., *RNA Biology*, 2020. doi: 10.1080/15476286.2020.1767951) and human (Brumwell et al., *RNA*, 2020, doi: 10.1261/rna.070318.119.).

We added these references.

26) Are the snoRNAs of Ψ 530 and Ψ 522 differentially expressed between BSF and PCF in a way that could explain the difference in Ψ levels? Would overexpression of *TB11Cs6H1* in BSF increase levels of Ψ 530 to those of PCF and would that results in change in *eS12* presence in the ribosomes?

The snoRNA guiding Ψ 530 is decreased, whereas Ψ 522 is increased in BSF compared to PCF. Ψ 530 is not developmentally regulated. The effect on *eS12* binding may stem from the effect on rRNA structure elicited by the reduction in Ψ 530.

27) Please indicate on which fraction of RNA was DMS-MaPSeq conducted (polysome/monosome etc).

The experiment was done *in vivo*, and DMS-MaPSeq was conducted on total RNA. However, this is no longer relevant because we removed this experiment. DMS mapping in indirect and the cryo-EM better demonstrates the effect on ribosome structure.

Reviewer #3 (Remarks to the Author):

The manuscript by Rajan et al describes application of three RNA-Seq based methods to identify sites of pseudouridine modification in ribosomal RNAs of parasitic protozoan *Trypanosoma brucei*. The quality of these datasets is high and the most prominent hypo or hyper modified positions are largely consistent among approaches that detect chemical modification, stand breakdown or pausing at pseudouridine site. The focus then shifts to a specific position 530 and the *TB11Cs6H1* snoRNA, previously characterized by the same group, that presumably guides modification at this position. It is well established that PTC is heavily modified in probably all organisms, which brings at least two-fold burden on

establishing a critical role of any given position. Developmental changes in modification prevalence are indicative of biological relevance but a formal connection from particular snoRNA to modification site to (presumed) ribosome structural change to preferential translation of some mRNAs is an endeavor that requires efficient genetic tools to start with. Here, snoRNA connection to modification position rests on a single allele KO and the effects are borderline. If conditional snoRNA expression is possible, there is no apparent reason for not establishing a conditional null KO, be it by CRISPR or recombination. This would allow monitoring effects in a time-resolved manner and hopefully generate more apparent impact on selective mRNA translation or ribosome composition, or any other aspect of trypanosome metabolism. Limiting the observed phenotype to a single LSU modification is problematic considering the report of snRNA modifications by TB11Cs6H1 snoRNA. The conjecture between a single modification loss in LSU and S12 reduction is unclear just like the reasons for snoRNA sKO compensation by S12 overexpression. These are interesting preliminary findings on their own but give limited mechanistic insights at this point. Certainly worth pursuing further.

The *T. brucei* system is somewhat limited, and so far, no conditional KO is available for non-coding RNAs. In fact, we were unable to independently express the four snoRNAs for an unknown reason. We only managed to co-express combinations of three snoRNAs at a time. To the best of our knowledge, we are the only group that has managed to overexpress any non-coding RNA in trypanosomes! This task is easier in the related *Leishmania* species. As explained in the text, the change in Ψ 530 affected the ribosome structure, which resulted in the dislodging of eS12. Increasing its level by add-back provides a greater chance for it to bind to the ribosome. Indeed, hnRNPF/H levels were restored as a result of eS12 overexpression. We now discuss the possibility that not all the changes in the level of proteins emerge from the effect on the ribosomes, as this snoRNA also has a guiding function on snRNA. It seems that trypanosome snoRNAs can also act as anti-sense regulators, as demonstrated recently in our collaborative study (Guegan et al., *Sc. Adv.* 2022).

We hope that the extensive revision we conducted to address the reviewers' concerns, and especially providing the cryo-EM structure of the parental ribosomes and the mutant lacking a single Ψ , makes our study suitable for publication in *Nat. comm.* We look forward to hearing from you.

Sincerely yours,

Shulamit Michaeli
Professor,
Corresponding author

REVIEWER COMMENTS

Reviewer #1 (Remarks to the Author):

The authors have sufficiently addressed my comments. Thank you for the explanations to help clarify the manuscript.

Reviewer #2 (Remarks to the Author):

I would like to commend the authors for their detailed response and for including additional experiments that help strengthen some of their initial results. Major additions to the current version include 1) cryoEM structures of WT and sKO ribosomes, which suggest a partial change in the structure of the two ribosomes, 2) quantification of pseudouridine via tandem LC-MS, as an additional validation method, and 3) overexpress the snoRNAs regulating modification of H69.

However, despite these additions, some of my major concerns remain unresolved. I find this work, although addressing very interesting and important questions, to be limited in its mechanistic insights and in its ability to connect between the modification of a specific site in rRNA and between translation. First, Although LC-MS was added, and in some cases reinforces the conclusions from profiling with other methods, in the case of the debatable sites of interest (e.g., 522, 524, 528, 530, which are in helix 69 and two of them are genetically manipulated in this study) LC-MS was not able to reconcile the discrepancy, since no data was obtained for these sites in the LC-MS analysis. Thus, the changes in modifications of these sites are still not sufficiently validated.

Second, as mentioned before, the global profiling of the two life stages seems disconnected from the rest of the experiments (i.e., focusing on sites in H69 and their functional roles), and seems to be better presented as two individual papers (as also hinted by the authors in their response).

Third, I was not convinced that hnRNP F/H is an appropriate reporter to be used in order to prove the effect of the sKO on protein translation, and found the evidence of how this gene is affected by the sKO (and addback experiments) not sufficient to infer the proposed mechanistic insights. This is further detailed below in my specific comments.

Forth, this reviewer appreciates the fact that knock-down (or inducible knock-down) of these snoRNAs is challenging. However, the strategy of overexpressing the snoRNAs regulating H69 sites seems problematic, as most of these sites are modified at 100% stoichiometry, and thus any effect seen by snoRNA overexpression might be caused by other mechanisms, rather than by modulating the level of pseudouridine in those sites.

Below I detail my thoughts regarding authors' response to my comments (I focused only on comments where I felt authors reply did not satisfy my concern), and regarding the new data that was added since I last saw the manuscript.

I hope the authors find these comments useful.

Specific thoughts regarding my comments from the first round of review (numbering according to my

original major comment):

- 1) My original comment still stands. Although authors show a comprehensive examination of the modified sites in the rRNA (and even add LC-MS) this part is strictly descriptive, and does not relate strongly to the rest of the experiments in the study. In addition, the overlap between methods is sometimes very limited and emphasizes that the catalog they generated requires additional validation (as they also comment in the revised text). Of note, most sites in H69 were not analysed by LC-MS.
- 3) P-values have been added, but again without mentioning whether correction for multiple hypothesis was done (as also requested by Reviewer #1). Also, authors write throughout the manuscript that they regard only p -value < 0.05 when in fact they relate to $\Psi 518$ as being significant although it has $p > 0.05$. In addition, authors didn't reply as to how nanopore data was analysed (doesn't appear at all in methods section) and how significance was assigned.
- 4) Authors managed to validate mostly sites which aren't relevant beyond the initial profiling (i.e., they are not in H69 and are not examined in further experiments in the paper). For sites in helix H69, on which they focus mechanistically, they did not resolve the discrepancy between methods. Also, $\Psi 518$ was validated by 2 methods, not 3 as they state, since results in HydraSeq didn't meet the $p < 0.05$ criteria.
- 6) The existing panel is problematic. Authors try to correlate between expression of snoRNA TB11Cs6H1 (mediating $\Psi 530$), the presence of eS12 on 80S monosomes and protein levels of hnRNP F/H. Originally, they claim loss of snoRNA causes reduction in hnRNP's levels (Fig. 2H and Fig. 4Dii only 2 left columns). However, panels of eS12 addback (Fig. 4Dii, two right columns) suggest that levels of hnRNP are higher in the sKO cells. This discrepancy isn't reconciled and thus it is problematic to draw conclusions as to the effect of the manipulations on the level of hnRNP F/H (one of the major claims of the paper). Again, it is suggested authors validate effects of sKO and eS12 on more than just one protein target, preferably on more robust targets found also in the MS (Fig. 2G).
- 8) In light of comments above, results on expression of hnRNP F/H aren't solid enough and additional proteins should be shown to validate the hypothesized connection between $\Psi 530$, eS12 and regulation of levels of specific proteins.
- 9) According to the authors' reply, abundance of hnRNP F/H is too low for it to be detected in the MS. Since they focus only on hnRNP F/H which did not come up in the MS, it isn't clear what the additive value of the MS is to the study. Furthermore, since hnRNP F/H was the only validated protein (by WB) this means that none of the changing proteins identified in the MS (Fig. 2G) were validated. Authors should validate changing proteins from the MS and show that they are affected by the sKO, otherwise there's no relevance to their MS and the choice of hnRNP F/H is arbitrary (due to prior papers/finding but unrelated to the story they tell to this point).
- 10) It is commendable that authors incorporated CryoEM analysis to their work, especially since these results confirm their statement that loss of $\Psi 530$ results in structural changes. However, since authors use this to further show differential localization of eS12 in WT and sKO ribosomes it would be valuable to indicate which additional proteins change their interaction with the ribosomes in these samples (also in light of Fig. 4B).

Additional specific comments on the newly added data:

- 1) Line 137-139: "This method... PCF", should add "but failed to reconcile the effect on Ψ 522 and Ψ 524 due to lack of information on these sites".
- 2) Line 150-151: change to "while SSU_ Ψ 1088 was identified as a hypermodified site ONLY by...".
- 3) Line 154 (and others, including some figures): Authors relate to Ψ 518 as significant although it isn't.
- 4) Line 165/169: Authors relate to Ψ 1166 as significant although it isn't.
- 5) Line 176: writing is misleading as most sites in H69 had inconsistent results across platforms.
- 6) Line 245: it would strengthen the authors' claim if they could mention that most proteins found to be affected by addback were also affected by sKO, especially if the direction of effect fits.
- 7) Table 7: what is "1161_OE"?
- 8) Line 270 (Fig. S8): it isn't clear what is M1, M2, M3, 4. Which mutant is which? Is WT the overexpression of 4 WT snoRNAs or of no snoRNAs at all? Also, sKO of Cs6H1 affected levels of hnRNP F/H but M1 doesn't show this effect, although in this mutant the relevant snoRNAs' levels are reduced.
- 9) Line 266-272: Authors mention that overexpression of mutants should eliminate expression of snoRNAs, but show elevation of levels in Ms vs WT.
- 10) Line 272: Authors state "No growth defects..." although it seems the slopes of the strains is different. This should be repeated with more replicates per strain and slopes should be quantified to show differences (or not) in growth rates.
- 11) Authors chose to overexpress the snoRNAs. However, with the exception of Ψ 518, all Ψ s affected by these snoRNAs are at 100% stoichiometry (according to tandem LC-MS). So, overexpressing the snoRNAs should have little effect of modification level of these sites, and any observed effects might be due to other mechanisms.

Reviewer #3 (Remarks to the Author):

The revised submission addressed some of my initial concerns while additional experiments dramatically expanded the scope of the paper. In doing so, more questions have been raised inserted of resolving those brought up in the first round of review. Addition of new high resolution ribosome structures would be a major boon for the field. However, these structures show loss of S12 and introduce more ambiguity to the mechanism of "specific effect of rRNA modification on translation." Indeed, new data make direct nature of this conclusions less than apparent. At the high level, the tour de force experiments are remarkably well executed, but add little to a establishing a mechanistic link between one allele KO for a snoRNA and loss of pseudouridylation in Helix 69 in LSU, and, now, to the loss of S12 from the small subunit.

Below are some suggestions on how to improve the content delivery and, ideally, decrease the number of twists and turns in the article.

Abstract: too many undefined abbreviations (HydraPsiSeq, LC-MS, H69). "First" is arbitrary, based on author's knowledge at this time, and can change any time.

Line 48: "heterogeneity of ribosomal proteins (RP) (existence of paralogs) as well as their stoichiometry and modification" should this read "due to existence of paralogs?"

Line 55: "less than half of the modified sites are sub-stoichiometric" – this is ambiguous.

Line 56: "However" – there seems to be no contrast with something that had been said before.

Line 78: Define what is "H69" and why it is important.

Line 99: remove "additional"

Line 157: "proposed" sites perhaps, not structure.

Line 158: insert ref to ribosome structure.

Line 187: "solitary single copy gene" seems redundant and confusing in the case of perpetually diploid organism.

Line 218: The authors state that hnRNPF/H can be used as read-out for the effect of sKO on translation.

Could it be that hnRNPF/H binds snoRNA and the loss is a direct result of snoRNA depletion? Does depletion of hnRNPF/H affect the snoRNA?

Lines 237-241. Inserting five sub-panels with different types of data into a single lettered panel is confusing to a reader. Suggest removing i ii iii ...and braking down the panel to keeping one type of data per lettered panel. Same applies to other figures.

Line 262. Suggest eliminating this entire chapter. This is a negative result that does not support the conclusion stated in Lane 280 and distract the reader from main message, which is not easy to follow as it is.

Thank you for giving us the opportunity to further revise our manuscript. As suggested by you and the reviewers, we toned down our claim regarding translation regulation and decided to change the title accordingly to "*A single pseudouridine on rRNA matters: A specific effect of rRNA modification on ribosome structure and function in the mammalian parasite Trypanosoma brucei*".

Over the past year we conducted additional experiments that further support our initial claims but did not add these results to the previous revision, because we wished to limit our changes to those that addressed the reviewers' comments. These data provide mechanistic insight to explain the defects seen upon knock-out of the single pseudouridine. These results are now included in the current revised manuscript to address the additional concerns raised by one of the reviewers.

Below please find our detailed responses to the reviewers' comments.

Reviewer #1

We were very gratified that this reviewer appreciated our revision.

Reviewer #2

We were happy to see that this reviewer found the revision satisfying in many respects, as he/she outlined. However, additional concerns were raised, especially regarding mechanistic insights, and the connection between the altered modification and translation. To address this issue, we now provide evidence that the 3' UTR of the mRNAs whose translation is affected in the sKO cells, is crucial for translational regulation. We provide an additional example of aldolase that was shown to be reduced by proteome analysis in the sKO cells due to its regulation by 3' UTR. These data also support the observation that the effect on translation is not due to unique characteristics of the CDS (i.e., codon usage). We have not yet identified the specific 3' UTR sequence that dictates

this specific effect. Accordingly, we toned down our claim regarding the precise mechanism.

Please find below our detailed answers to the reviewers' specific comments.

A) Although LC-MS was added, and in some cases reinforces the conclusions from profiling with other methods, in the case of the debatable sites of interest (e.g., 522, 524, 528, 530, which are in helix 69 and two of them are genetically manipulated in this study) LC-MS was not able to reconcile the discrepancy, since no data was obtained for these sites in the LC-MS analysis. Thus, the changes in modifications of these sites are still not sufficiently validated.

The reviewer is correct, but the LC-MS data does support the developmental regulation of these sites. The main Ψ we studied, 530 was not claimed to be developmentally regulated in total RNA (PCF versus BSF) but was found to be hypermodified in PCF polysomes. It is well known that H69 positions are not regulated in other systems, but as discussed in the manuscript, are essential for ribosome function in species ranging from bacteria to yeast, and this is why we decided to study their function in these parasites. Our study strikingly shows that in contrast to other systems, even a single Ψ matters, and can affect ribosome structure and function.

B) Second, as mentioned before, the global profiling of the two life stages seems disconnected from the rest of the experiments (i.e., focusing on sites in H69 and their functional roles), and seems to be better presented as two individual papers (as also hinted by the authors in their response).

The reviewer is correct that our study has two separate parts. However, the information regarding the differential modifications and comparison between the four methods is essential these days to evaluate the potential of such changes to affect ribosome function. This is why it is very timely to incorporate the data to this study. The study supports the existence of most Ψ s in all methods. In addition, the snoRNA guiding these differentially regulated Ψ s are not amenable to genetic manipulation since these snoRNAs guiding these positions are present in polycistronic snoRNA clusters, and these are repeated several times in the genome. RNAi studies in our lab showed that silencing H/ACA

snoRNAs is not efficient enough to observe a phenotype. This is why we provide interesting results only on solitary snoRNAs.

The developmentally regulated sites are not located in functional domains of the ribosome, but we chose to study the snoRNA guiding Ψ on H69 because Ψ 530 was preferentially found in the polysomal fraction and in addition, Ψ s on H69 were studied in other organisms. We suspected the very special function of these sites in parasites, because in contrast to most snoRNAs, these snoRNAs guiding on H69 are present in single copy genes. Therefore, the function of these sites can be studied by eliminating its Ψ using CRISPR-Cas9 technology. Note that the snoRNAs guiding these modifications, especially Ψ 530 were not yet identified in humans!!

As suggested by the reviewer, we toned down the claims regarding the differential regulation between the two life stages, although these do exist.

We wish to emphasize that we recently also determined the level of Ψ by HydraPsiSeq of slender (BSF from infected animals) and stumpy forms of the parasites obtained from animals, and observed similar developmental regulation as presented in the manuscript. However, we did not observe any change in Ψ 530 and accordingly, did not add these results to the revised manuscript. Attached below is the link for the reviewer to access these data.

https://weizmannacil-my.sharepoint.com/:x:/g/person/shanmugha-rajan_k_weizmann_ac_il/EWDH3HdlvMdEiBSqlquBZd0BCC6dvnA-6D7qFDlhTvewdA?e=9ejUHT

Interestingly, in the related parasite, *Leishmania* we also observed developmental regulation of the Ψ 530 homologue and showed that its level differs between parasites causing visceral versus cutaneous disease.

Combining the two parts of the paper emphasizes the significance of sites that are not developmentally regulated for the normal function of the ribosomes and that regulation on such housekeeping sites cannot be tolerated by the parasites.

We think that it is important to demonstrate the overall differential modification phenomena, while also demonstrating the effect of Ψ at important positions. This is why we think that two aspects of the study complement each other.

C) Third, I was not convinced that hnRNP F/H is an appropriate reporter to be used in order to prove the effect of the sKO on protein translation, and found the evidence of how this gene is affected by the sKO (and addback experiments) not sufficient to infer the proposed mechanistic insights. This is further detailed below in my specific comments.

The study on hnRNPF/H began when we screened antibodies prepared in our group to see if any of the RNA binding proteins that we studied are affected in the sKO. To our surprise hnRNPF/H was found to be severely affected. Unfortunately, this protein is not sufficiently abundant to be detected in the PCF proteome. Because it was so strongly affected, we decided to study the mechanism by which it is regulated in the KO cells. This led us to replace the 3' UTR of the hnRNPF/H by *in situ* tagging. The results clearly showed that the regulatory sequence lies in the 3' UTR region. Note that we also replaced the 5' UTR but we could not observe any effect in the sKO cells and thus decided not to include in the manuscript. To address another concern, that the protein we studied was not present in the list obtained from the proteome analysis, we now provide data regarding aldolase, demonstrating that the effect of the sKO is also exerted through the 3' UTR.

D) Forth, this reviewer appreciates the fact that knock-down (or inducible knock-down) of these snoRNAs is challenging. However, the strategy of overexpressing the snoRNAs regulating H69 sites seems problematic, as most of these sites are modified at 100% stoichiometry, and thus any effect seen by snoRNA overexpression might be caused by other mechanisms, rather than by modulating the level of pseudouridine in those sites.

In fact, both reviewers were uncomfortable with this strategy, which is the only possible approach, so we decided to remove this section, as suggested by the third reviewer.

Below I detail my thoughts regarding authors' response to my comments (I focused only on comments where I felt authors reply did not satisfy my concern), and regarding the new data that was added since I last saw the manuscript. I hope the authors find these comments useful.

We thank the reviewer for all the valuable comments, which we addressed below.

Specific thoughts regarding my comments from the first round of review (numbering according to my original major comment):

1) My original comment still stands. Although authors show a comprehensive examination of the modified sites in the rRNA (and even add LC-MS) this part is strictly descriptive, and does not relate strongly to the rest of the experiments in the study. In addition, the overlap between methods is sometimes very limited and emphasizes that the catalog they generated requires additional validation (as they also comment in the revised text). Of note, most sites in H69 were not analysed by LC-MS.

We nevertheless think this information, comparing all the available methods for genome-wide mapping of Ψ , is of utmost importance for the Ψ modification field. We already mentioned (see above) our arguments explaining why, despite the fact that H69 Ψ s are unchanged between the life stages, it is important to study their biological role. In fact, the LC-MS detected and quantified the H69 Ψ sites, but did not reveal differential regulation.

3) P-values have been added, but again without mentioning whether correction for multiple hypothesis was done (as also requested by Reviewer #1).

We calculated the corrected p -value and fold-change using DESeq2 (in which the p -values are corrected for multiple testing using the Benjamini and Hochberg method) for RNA-seq, and for the proteome data we used Benjamini-Hochberg correction for multiple hypothesis testing (SignificanceB), and this is now mentioned in Materials and Methods. For the modification levels, since the fold change differs among the methods, this analysis is not useful because it could diminish the effect.

Also, authors write throughout the manuscript that they regard only p -value < 0.05 when in fact they relate to $\Psi518$ as being significant although it has $p > 0.05$.

We accept this point and removed analysis of this site from the current version.

In addition, authors didn't reply as to how nanopore data was analysed (doesn't appear at all in methods section) and how significance was assigned.

We now added a description of the data analysis to the revised version.

4) Authors managed to validate mostly sites which aren't relevant beyond the initial profiling (i.e., they are not in H69 and are not examined in further experiments in the paper).

Identification of such sites is important on its own because it shows the validity of the different methods and further supports the concept of developmental regulation. We continue to work on the other developmentally regulated sites, but it will take time and there is no guarantee that a phenotype will be observed.

For sites in helix H69, on which they focus mechanistically, they did not resolve the discrepancy between methods.

This is true. One of the reasons might be that these sites are close to each other, which might be problematic for quantification. The significance of these sites is not due to their developmental regulation, but rather, because of their important location.

Also, Ψ 518 was validated by 2 methods, not 3 as they state, since results in HydraSeq didn't meet the $p < 0.05$ criteria.

We agree, and the discussion of this site was removed.

6) The existing panel is problematic. Authors try to correlate between expression of snoRNA TB11Cs6H1 (mediating Ψ 530), the presence of eS12 on 80S monosomes and protein levels of hnRNP F/H. Originally, they claim loss of snoRNA causes reduction in hnRNP's levels (Fig. 2H and Fig. 4Dii only 2 left columns). However, panels of eS12 addback (Fig. 4Dii, two right columns) suggest that levels of hnRNP are higher in the sKO cells. This discrepancy isn't reconciled and thus it is problematic to draw conclusions as to the effect of the manipulations on the level of hnRNP F/H (one of the major claims of the paper).

The comment is not clear to us. We wrote in the text that overexpression of eS12 in wild type generated a defect. However, in sKO we specifically show that the defect was reversed, as overexpression in the absence of eS12 background resulted in normal levels of hnRNPF/H expression. To strengthen support for this concept and as requested by the

reviewer, we added data regarding aldolase, showing that its translation is dependent on eS12.

Again, it is suggested authors validate effects of sKO and eS12 on more than just one protein target, preferably on more robust targets found also in the MS (Fig. 2G).

As mentioned above, we added data regarding aldolase.

8) In light of comments above, results on expression of hnRNP F/H aren't solid enough and additional proteins should be shown to validate the hypothesized connection between Ψ 530, eS12 and regulation of levels of specific proteins.

We added data regarding aldolase, and also demonstrated that the effect on hnRNPF/H expression in the sKO depends on its 3' UTR.

9) According to the authors' reply, abundance of hnRNP F/H is too low for it to be detected in the MS. Since they focus only on hnRNP F/H which did not come up in the MS, it isn't clear what the additive value of the MS is to the study. Furthermore, since hnRNP F/H was the only validated protein (by WB) this means that none of the changing proteins identified in the MS (Fig. 2G) were validated. Authors should validate changing proteins from the MS and show that they are affected by the sKO, otherwise there's no relevance to their MS and the choice of hnRNP F/H is arbitrary (due to prior papers/finding but unrelated to the story they tell to this point).

The hnRNPF/H experiment is a very good example to show the effect of snoRNA sKO on protein level, but its expression is very low in this life stage (PCF). We now provide data from aldolase in which the mRNA level was unchanged in the sKO, but its level was decreased in the proteome of the sKO.

10) It is commendable that authors incorporated CryoEM analysis to their work, especially since these results confirm their statement that loss of Ψ 530 results in structural changes. However, since authors use this to further show differential localization of eS12 in WT and sKO ribosomes it would be valuable to indicate which additional proteins change their interaction with the ribosomes in these samples (also in light of Fig. 4B).

Careful analysis of all RPs in the Cryo-EM structure showed that only eS12 was absent in the sKO, and all other proteins were present in the structure as well as in the MS data. The full dataset is presented in Supplementary Table 9.

Additional specific comments on the newly added data:

1) Line 137-139: *“This method... PCF”, should add “but failed to reconcile the effect on Ψ 522 and Ψ 524 due to lack of information on these sites”.*

We added the suggested text.

2) Line 150-151: *change to “while SSU_ Ψ 1088 was identified as a hypermodified site ONLY by...”.*

"only" added.

3) Line 154 (and others, including some figures): *Authors relate to Ψ 518 as significant although it isn't.*

This statement was removed from the manuscript, as indicated above.

4) Line 165/169: *Authors relate to Ψ 1166 as significant although it isn't.*

We removed this assertion.

5) Line 176: *writing is misleading as most sites in H69 had inconsistent results across platforms.*

We changed the statement and agreed that was not presented accurately.

6) Line 245: *it would strengthen the authors' claim if they could mention that most proteins found to be affected by addback were also affected by sKO, especially if the direction of effect fits.*

The data do not allow us to make this claim since there is no tight correlation between the results obtained by overexpression and sKO. This can be explained by the secondary effect on the snoRNA population due to competition for snoRNP proteins by the overexpressing snoRNA. We decided to remove these data from the paper because the

results are confusing and do not contribute to the understanding of the effect of this snoRNA on translation.

7) *Table 7: what is “1161_OE”?*

As stated above, we removed these data.

8) *Line 270 (Fig. S8): it isn't clear what is M1, M2, M3, 4. Which mutant is which? Is WT the overexpression of 4 WT snoRNAs or of no snoRNAs at all? Also, sKO of Cs6H1 affected levels of hnRNP F/H but M1 doesn't show this effect, although in this mutant the relevant snoRNAs' levels are reduced.*

All these data were removed as suggested by Reviewers 2 and 3.

9) *Line 266-272: Authors mention that overexpression of mutants should eliminate expression of snoRNAs, but show elevation of levels in Ms vs WT.*

In each case, one snoRNA was mutated and hence not expressed, while the rest were overexpressed. However, this is no longer relevant, because these experiments were removed based on the reviewer's recommendation.

10) *Line 272: Authors state “No growth defects...” although it seems the slopes of the strains is different. This should be repeated with more replicates per strain and slopes should be quantified to show differences (or not) in growth rates.*

As mentioned above, all these results were removed.

11) *Authors chose to overexpress the snoRNAs. However, with the exception of Ψ 518, all Ψ s affected by these snoRNAs are at 100% stoichiometry (according to tandem LC-MS). So, overexpressing the snoRNAs should have little effect of modification level of these sites, and any observed affects might be due to other mechanisms.*

The reviewer is right. Anyway, these data were removed.

Reviewer #3 (Remarks to the Author):

The revised submission addressed some of my initial concerns while additional experiments dramatically expanded the scope of the paper. In doing so, more questions have been raised instead of resolving those brought up in the first round of review. Addition of new high resolution ribosome structures would be a major boon for the field. However, these structures show loss of S12 and introduce more ambiguity to the mechanism of “specific effect of rRNA modification on translation.” Indeed, new data make direct nature of this conclusions less than apparent. At the high level, the four de force experiments are remarkably well executed, but add little to establishing a mechanistic link between one allele KO for a snoRNA and loss of pseudouridylation in Helix 69 in LSU, and, now, to the loss of S12 from the small subunit. Below are some suggestions on how to improve the content delivery and, ideally, decrease the number of twists and turns in the article.

We thank this reviewer for appreciating the efforts we invested in the revision and in providing additional data. As explained in our previous submission, overexpression of multiple snoRNAs was the only way we could functionally address the importance of the H/ACA snoRNA guiding Ψ positions on H69. Based on this reviewer's recommendation and the comments of the second reviewer, we removed these results in the revised manuscript.

Abstract: too many undefined abbreviations (HydraPsiSeq, LC-MS, H69). “First” is arbitrary, based on author’s knowledge at this time, and can change any time.

We revised the Abstract based on this comment and eliminated many of the abbreviations.

Line 48: *“heterogeneity of ribosomal proteins (RP) (existence of paralogs) as well as their stoichiometry and modification” should this read “due to existence of paralogs?”*

This mistake was corrected.

Line 55: “less than half of the modified sites are sub-stoichiometric” – this is ambiguous. We changed the sentence to “50% of the modified sites are only partially modified”.

Line 56: “However” – there seems to be no contrast with something that had been said before.

The ambiguity was corrected.

Line 78: Define what is “H69” and why it is important.

The domain is denoted as helix 69 in all publications, and we followed this conventional nomenclature. Its importance was described throughout the paper, but we now added more information to the Introduction.

Line 99: remove “additional”

Removed.

Line 157: “proposed” sites perhaps, not structure.

We changed this to “cryo-EM structure”.

Line 158: insert ref to ribosome structure.

We inserted the requested reference.

Line 187: *“solitary single copy gene” seems redundant and confusing in the case of perpetually diploid organism.*

The term differentiates these genes from most of the trypanosome snoRNA genes, which are located in clusters carrying multiple snoRNAs that are also reiterated several times in the genome. Nevertheless, we changed this sentence to "solitary gene encoding for a single snoRNA".

Line 218: The authors state that hnRNPF/H can be used as read-out for the effect of sKO on translation. Could it be that hnRNPF/H binds snoRNA and the loss is a direct result of snoRNA depletion? Does depletion of hnRNPF/H affect the snoRNA?

This aspect was not studied, although we see no effect on hnRNPF/H mRNA level following TB11Cs6H1 sKO. We did not examine the effect of hnRNPF/H depletion on TB11Cs6H1 because we did not see a reason for doing this experiment.

Lines 237-241. Inserting five sub-panels with different types of data into a single lettered panel is confusing to a reader. Suggest removing i ii iii ...and braking down the panel to keeping one type of data per lettered panel applies to other figures.

We revised the figures and changed most of the sub panels to letter designations.

Line 262. Suggest eliminating this entire chapter. This is a negative result that does not support the conclusion stated in Lane 280 and distract the reader from main message, which is not easy to follow as it is.

We agree and removed this section based on both reviewers' suggestions.

We hope that this revision addresses the reviewers' remaining concerns, and makes our study suitable for publication in in *Nat. comm.* We look forward to hearing from you.

Sincerely yours,

Shulamit Michaeli

Professor,

Corresponding author

REVIEWERS' COMMENTS

Reviewer #2 (Remarks to the Author):

Changes made by the authors have improved the clarity of their work. The modifications introduced in this revision emphasize their actual findings, resulting in a more concise and coherent presentation of the research.

Reviewer #3 (Remarks to the Author):

I was satisfied by the 1st revision, and the second flushed out some issues noted by other reviewers.